# Contribution of different processes to changes in tropical lower stratospheric water vapor in chemistry-climate models

Kevin M. Smalley[1], Andrew E. Dessler[1], Slimane Bekki[2], Makoto Deushi[3], Marion Marchand[2], Olaf Morgenstern[4], David A. Plummer[5], Kiyotaka Shibata[6], Yousuke Yamashita[7,8], and Guang Zeng[4]

[1]Department of Atmospheric Science, Texas A&M, College Station, Texas, USA.
[2]LATMOS, Institut Pierre Simon Laplace (IPSL), Paris, France
[3]Meteorological Research Institute, 1-1 Nagamine, Tsukuba, Ibaraki 305-0052, Japan
[4]National Institute of Water and Atmospheric Research (NIWA), Wellington, New Zealand
[5]Canadian Centre for Climate Modelling and Analysis, Environment and Climate Change Canada
[6]School of Environmental Science and Engineering, Kochi University of Technology
[7]National institute for Environmental Studies (NIES)
[8]Now at Japan Agency for Marine-Earth Science and Technology (JAMSTEC), Yokohama, Japan

*Correspondence to:* Andrew Dessler (adessler@tamu.edu)

**Abstract.** Variations in tropical lower-stratospheric humidity influence both the chemistry and climate of the atmosphere. We analyze tropical lower stratospheric water vapor in $21^{st}$-century simulations from 12 state-of-the-art chemistry-climate models (CCMs), using a linear regression model to determine the factors driving the trends and variability. Within CCMs, warming of the troposphere primarily drives the long-term trend in stratospheric humidity. This is partially offset in most CCMs by an increase in the strength of the Brewer-Dobson circulation, which tends to cool the tropical tropopause layer (TTL). We also apply the regression model to individual decades from the $21^{st}$ century CCM runs and compare them to a regression of a decade of observations. Many of the CCMs, but not all, compare well with these observation, lending credibility to their predictions. One notable deficiency in most CCMs is that they underestimate the impact of the quasi-biennial oscillation on lower-stratospheric water vapor. Our analysis provides a new and potentially superior way to evaluate model trends in lower-stratospheric humidity.

## 1   Introduction

Stratospheric water vapor is well-known to be a greenhouse gas (e.g. Manabe and Wetherald, 1967; Forster and Shine, 1999; Solomon et al., 2010; Maycock et al., 2014). Because of this, understanding the processes that control the humidity of air entering the tropical lower stratosphere (hereafter $[H_2O]_{entry}$) has been a high priority of the scientific community since Brewer (1949) first described the stratospheric circulation.

It is now well established that the fundamental control over $[H_2O]_{entry}$ comes from the cold temperatures found in the tropical tropopause layer (TTL) (Fueglistaler et al., 2009b), and that variability in these temperatures translates into variability in $[H_2O]_{entry}$. The most well-known example of this is the so-called "tape recorder," in which the seasonal cycle in TTL temperatures is imprinted on tropical stratospheric water vapor (Mote et al., 1996).

On interannual time scales, variability in $[H_2O]_{entry}$ originates from variability in the Brewer-Dobson Circulation (BDC) variability (Randel et al., 2006a; Castanheira et al., 2012; Fueglistaler et al., 2014; Gilford et al., 2016) and the quasi-biennial oscillation (QBO) (O'Sullivan and Dunkerton, 1997; Randel et al., 1998; Dunkerton, 2001; Fueglistaler and Haynes, 2005; Choiu et al., 2006; Liang et al., 2011; Castanheira et al., 2012; Khosrawi et al., 2013; Kawatani et al., 2014; Tao et al., 2015).

Dessler et al. (2013, 2014) suggests that the temperature of the troposphere also exerts an influence on $[H_2O]_{entry}$ based primarily on an analysis of satellite measurements of $[H_2O]_{entry}$. This is mainly caused by radiative heating of the TTL from increased upwelling radiation from a warming troposphere (Lin et al., 2017). In addition to this mechanism, Dessler et al. (2016) demonstrated in two CCMs that a warming climate also increased the amount of water directly injected into the stratosphere via deep convection, providing another mechanism for tropospheric temperature to affect $[H_2O]_{entry}$.

Putting these factors together, Dessler et al. (2013, 2014) demonstrated that observed $[H_2O]_{entry}$ anomalies could be accurately reproduced with a simple linear model:

$$[H_2O]_{entry} = \beta_0 + \beta_{\Delta T}\Delta T + \beta_{BDC}BDC + \beta_{QBO}QBO + \epsilon \qquad (1)$$

Where $\Delta T$ is the temperature of the troposphere, BDC is the strength of the Brewer-Dobson circulation, QBO represents the phase of the QBO, and epsilon is the residual. Dessler et al. (2013) analyzed the $21^{st}$ century trend in one chemistry-climate

model (hereafter, CCM; they are similar to general circulation models, but with a more realistic stratosphere and higher vertical resolution in the TTL) and found that the regression model worked well in reproducing the CCM's $[H_2O]_{entry}$ trend over the $21^{st}$ century. They concluded that the increase in $[H_2O]_{entry}$ was driven by the increase in tropospheric temperatures, which was partially offset by a strengthening BDC.

Dessler et al. (2013)'s regression method provides a novel way to examine the regulation of $[H_2O]_{entry}$ in CCMs and

compare it to observations. The purpose of this paper is to see whether this linear decomposition of $[H_2O]_{entry}$ variability holds in most CCM and whether the same factors dominate.

## 2  Models

We analyze model output from 7 CCMs participating in Phase 2 of the Chemistry-Climate Model Validation Project (CCMVal-2) (Morgenstern et al. (2010); SPARC (2010)) and output from 5 CCMs participating in Phase 1 of the Chemistry-Climate

Model Initiative (CCMI-1) (Morgenstern et al. (2017)). Table 1 lists the model specifics and documentation.

We use simulations from the REF-B2 scenario of CCMVal-2. In this scenario, greenhouse gas concentrations during the $21^{st}$ century come from the A1B scenario, which lies in the middle of the SRES scenarios (IPCC, 2001). Ozone-depleting substances come from the halogen emission scenario A1 (WMO, 2007). CCMVal-2 specifics can be found in SPARC (2010) and Morgenstern et al. (2010). We use the refC2 scenario of the CCMI-1. In this scenario, greenhouse gas concentrations

come from the RCP 6.0 scenario (Meinshausen et al., 2011) and ozone-depleting substances come from the halogen emission scenario A1 (WMO, 2014). CCMI-1 model specifics can be found in Morgenstern et al. (2017). In order to maintain a consistent reference period between models, our analysis covers 2000-2097, which we will hereafter refer to as "the $21^{st}$ century ".

For each model, we fit CCM $[H_2O]_{entry}$ using the multivariate linear regression (MLR) model described above. We use tropical average 80-hPa water vapor volume mixing ratio as a proxy for $[H_2O]_{entry}$ (all tropical averages in this paper are averages over 30°N-30°S).

For our BDC index, we use 80-hPa diabatic heating rate (see Fueglistaler et al. (2009a) for details). Within models, studies have shown that the strength of the BDC increases throughout the 21st century, primarily resulting from increasing greenhouse gases (e.g. Austin and Li, 2006; Garcia and Randel, 2008; Li et al., 2008; Oman et al., 2008). Observations generally confirm that tropical upwelling into the lower stratosphere has strengthened (Bönisch et al., 2011; Randel and Thompson, 2011; Young et al., 2012; Sioris et al., 2014). However, the BDC is not a directly observable circulation, and different variables including: tracer gases, residual velocity, mean age of the air, and diabatic heating have been used (Rosenlof et al., 1997; Randel et al., 2006a; Okamoto et al., 2011; Seviour et al., 2012; Stiller et al., 2012). Thus, depending on the variable used, the strength of the connection between the BDC term and $[H_2O]_{entry}$ may change.

The tropospheric temperature index is the 500-hPa tropical average temperature. For the few CCMI-1 simulations that only produce variables on hybrid pressure levels (CMAM, CCSRNIES-MIROC3.2, and MRI-ESM1r1), we choose a hybrid pressure level close to the 500-hPa pressure surface (See Table 1). For the QBO index, we take the standardized anomaly of equatorial 50-hPa zonal winds (anomalies in this paper are calculated by subtracting the mean seasonal cycle). By examining $21^{st}$ century 50 hPa zonal winds (shown in supplement figures), we find that only 5 of the 12 models simulate a QBO (table 1). As a result, we do not expect the QBO to significantly impact $[H_2O]_{entry}$ in many of the models.

All of these choices are similar to those used by Dessler et al. (2013, 2014). The MLR returns the coefficients for each regressor in Equation 1, along with an uncertainty for each coefficient. Unless otherwise noted, we use 95%-confidence intervals in this paper. Autocorrelation in the residuals is accounted for in the uncertainties following Santer et al. (2000). Finally, we will illustrate results with the MRI model; figures showing results derived from the other models can be found in the supplement.

## 3  Century Analysis

We first analyze the long-term trend in $[H_2O]_{entry}$ over the $21^{st}$ century. To do this, we calculate annual average values of $[H_2O]_{entry}$ and perform a MLR against annual averages of the indices for BDC, QBO and $\Delta T$. For consistency, all annual average time series have had the 2000-2010 mean subtracted out. Most models simulate $[H_2O]_{entry}$ increasing during the $21^{st}$ century (Gettelman et al., 2010; Kim et al., 2013). However, recent observational studies have concluded that no significant historical trend in water vapor entering the lower stratosphere exists (Scherer et al., 2008; Hegglin et al., 2014; Dessler et al., 2014).

Figure 1 shows that the fits to most of the models generate adjusted $R^2$ values greater than 0.8. The NIWA-UKCA century MLR has the lowest adjusted R$^2$, with a value of approximately 0.6. Overall, this result confirms the result of Dessler et al. (2013) that the regression model does a good job job reproducing the models' $[H_2O]_{entry}$. Because we have left long-term trends in the time series, we will refer to this as the "trended analysis".

## 3.1 Detrended 21$^{st}$ Century

One concern with the trended analysis is that the $[H_2O]_{entry}$, BDC, and $\Delta T$ time series are all dominated by long-term trends. In such a case, an MLR may produce a high adjusted $R^2$ even if there is no actual relation between the variables. To eliminate the influence of long-term trends on adjusted $R^2$, we detrend each variable using a Fourier Transform filter (Donnelly, 2006) to remove long-term variability ($> 10$ years). We then use the MLR on the detrended $[H_2O]_{entry}$ and the detrended indices. Detrending by removing the long-term linear trend yields similar results.

Figure 1 shows the adjusted $R^2$ for the detrended calculation. For most of the models, the adjusted $R^2$ for the detrended MLR is moderately smaller than that for the trended one. This confirms that the long-term trends in the data tend to inflate the adjusted $R^2$, at least a bit. But we also confirm that the models' detrended $[H_2O]_{entry}$ is also well represented by the same linear model (Equation 1). Large differences do exist for some CCMs. For instance, the CCSRNIES trended century MLR captures approximately $90\%$ of the variance in $[H_2O]_{entry}$, while the detrended century MLR only explains about $40\%$ of detrended variance; the CNRM-CM5-3, NIWA-UKCA, and WACCM show something similar.

## 3.2 Physical Process Effects

The coefficients from the trended and detrended calculations are listed in Tables 2 and 3 respectively. The product of the regression coefficient and its index quantifies that process' impact on $[H_2O]_{entry}$. As an example, MRI $[H_2O]_{entry}$ increases by about 1.2 ppmv during the 21$^{st}$ century (Figure 2). The regression shows that this is the result of a large increase in $[H_2O]_{entry}$ due to $\Delta T$ increases ( 1.5 ppmv) that is offset by a strengthening BDC, which reduces $[H_2O]_{entry}$ by approximately 0.3 ppmv. The regression finds virtually no change in $[H_2O]_{entry}$ in response to the QBO.

Figure 3 shows that $[H_2O]_{entry}$ increases as $\Delta T$ increases in all models and that the $\Delta T$ regression coefficients are similar for both trended and detrended MLRs. The coefficient for individual models ranges from 0.1 to 0.6 ppmv K$^{-1}$, with an average of 0.32 ppmv K$^{-1}$ and a standard deviation of 0.15 ppmv K$^{-1}$. It is worth pointing out that the models can get the right answer for the wrong reason. For example, spurious diffusion of water vapor through the tropopause has been shown to be an issue in models (e.g. Gettelman et al., 2010; Hardiman et al., 2015). This may impact the relationship between $[H_2O]_{entry}$ and tropospheric warming, thereby biasing our results. However, Dessler et al. (2016) was able to accurately simulate the stratospheric trend in two CCMs using a diffusion-free trajectory model, showing that, in some models at least, this is not an issue.

This figure also shows that the BDC coefficient is generally negative, meaning that a strengthening BDC reduces $[H_2O]_{entry}$. This is consistent with previous research, which showed that a stronger BDC reduces TTL temperatures and lower-stratospheric water vapor (Randel et al., 2006a; Gilford et al., 2016). The coefficient for individual models ranges from -12. to 4.3 ppmv (K/Day)$^{-1}$, with an average of -3.55 ppmv (K/Day)$^{-1}$ and a standard deviation of 4.45 ppmv (K/Day)$^{-1}$. Two models (CNRM-CM5-3 and NIWA-UKCA) yield positive BDC coefficients, indicating potential problems with these models. And the magnitude of the MRI BDC coefficients are about two times larger than those produced by MRI-ESM1r1. This could explain why the detrended adjusted $R^2$ value for MRI-ESM1r1 is so much smaller than that of MRI.

Figure 3 shows that all QBO regression coefficients are small, generally within $\pm$ 0.04 ppmv, with even the sign of the effect in doubt. Interestingly, one of the CCMs not simulating a QBO, CMAM-CCMI, produces the largest QBO regression coefficients of 0.082 $\pm$0.04 and 0.077 $\pm$0.04 ppmv for the trended and detrended calculations, respectively. Among CCMs that do simulate a QBO, the ensemble average QBO regression coefficient does not differ much from the same quantity (approximately 0 ppmv) for the other models. We will discuss this further in the next section.

As can be seen in the plots for individual models in the supplement, the variability in $[H_2O]_{entry}$ in a few models comes almost entirely from the variability in BDC, with almost no variability in the $\Delta T$ time series (other than the long-term trend). That means that the $\Delta T$ term, which is almost a pure trend, will fit whatever is left after matching the interannual variability and trend of the QBO time series.

We have also calculated the long-term linear trend of $[H_2O]_{entry}$ for each model, as well as the trend in each component of $[H_2O]_{entry}$, as determined by the multivariate fit (e.g., the trend in the components plotted in Fig. 2). We find that $\Delta T$ makes the largest contribution to the trend in $[H_2O]_{entry}$, with a smaller negative effect from the a strengthening BDC on $[H_2O]_{entry}$, and a trend of close to zero for the QBO (Figure 4).

To provide additional information about the relative contribution from the individual terms in eq. 1, we have also calculated the regression coefficient using standardized variables. To do this, we take each regression coefficient and multiply it by the standard deviation of the associated regressor index. The values are listed in tables 2 and 3 and they confirm that, in the trended calculations, $\Delta T$ is the dominant cause of the trend in $[H_2O]_{entry}$. The BDC acts to reduce the trend, but its overall impact is much smaller than $\Delta T$.

In the detrended calculations, the standardized $\Delta T$ regression coefficients are smaller than those from the trended calculations, while the magnitude of the BDC coefficients remains relatively constant. This results in the BDC being more important than $\Delta T$ for short-term variability. In all of our calculations, we find that the QBO has little impact on $[H_2O]_{entry}$.

## 4 Decadal Analysis

Ideally, we would compare the results of the last section to observations. Unfortunately, we don't have 100 years of observations to test the models against. Instead, we will compare regressions of 10-year segments from the CCMs to regressions of 10-years of observations. This will help us evaluate how good the models are and provide us with an indication of how representative a single decade is.

To do this, we split the $21^{st}$ century of each CCM run into 10 decades (2000-2010, 2010-2020, 2020-2030, 2040-2050, etc.) and fit each individual decade using the regression model (Equation 1). The regression calculation used on each 10-year segment is identical to the century analysis, except monthly averaged anomalies of all quantities are used instead of annual mean anomalies. Following Dessler et al. (2014), decadal regression terms are lagged in order to maximize MLR fit: we lag $\Delta T$ by 3 months, the BDC by 1 month, and the QBO by 3 months. These lags reflect the time between changes in each index and the impact on $[H_2O]_{entry}$.

Figure 5 shows the median ± one standard deviation of the ten decadal adjusted $R^2$ values generated by each CCM. The ensemble average is 0.61±0.25, with some spread among the models. Also plotted are the adjusted $R^2$ from two regressions of the tropical average Aura Microwave Limb Sounder (MLS) 82-hPa water vapor mixing ratio observations from Dessler et al. (2014). One regression uses Modern-Era Retrospective Analysis for Research and Applications reanalysis (MERRA) (Rienecker et al., 2011) and the other uses European Centre for Medium-Range Weather Forecasts interim reanalysis (ERAI) (Dee et al., 2011) for the $\Delta T$ and BDC indices; the QBO index is standardized anomaly of monthly and zonally averaged equatorial 50-hPa winds obtained from the NOAA Climate Prediction Center (http://www.cpc.ncep.noaa.gov/data/indices). The MLS data covers the time period 2004-2014.

Many of the models have a range of adjusted $R^2$ values that overlap with the observational regression. However, the models producing the smallest decadal adjusted $R^2$ values, CCSRNIES CNRM-CM5-3, and NIWA-UKCA, are also the models that produced the poorest fits to long-term detrended $[H_2O]_{entry}$. This provides some evidence that analysis of just a decade of $[H_2O]_{entry}$ can provide insight into the long-term behavior of that quantity.

Figure 6 shows the median and one standard deviation of each coefficient (values are listed in table 4), along with the coefficients from the regression of the MLS data (taken from Table 1 of Dessler et al. (2014)). We find that the CCMs agree unanimously that increases in $\Delta T$ are associated with increased $[H_2O]_{entry}$, though the CCM ensemble tends to underestimate the observational estimate. The only models that don't fall within both observational ranges are CCSRNIES, CMAM-CCMI, and CNRM-CM5-3.

In addition, the spread between the different decades for a single model tends to be small. The coefficient for individual models ranges from 0.01 to 0.4 ppmv K$^{-1}$, with an average of 0.15 ppmv K$^{-1}$ and a standard deviation of 0.11 ppmv K$^{-1}$. This provides additional confidence that the comparison between the CCMs and one decade of observations is meaningful.

Figure 6 shows that there exists significant spread in the CCMs' decadal BDC regression coefficients. The coefficient for individual models ranges from -8.4 to 2.9 ppmv (K/Day)$^{-1}$, with an average of -3.55 ppmv (K/Day)$^{-1}$ and a standard deviation of 3.58 ppmv (K/Day)$^{-1}$. On all timescales, we expect a strengthening BDC should cool the TTL and reduce $[H_2O]_{entry}$, so the coefficient should be negative. We see that the median is indeed negative for all CCMs except for the CNRM-CM5-3 and NIWA-UKCA (these models also generated positive BDC coefficients for the century analysis).

Comparing to observations, we find that the model ensemble does well. The CCSRNIES, CCSRNIES-MIROC-3.2, CMAM, CMAM-CCMI, LMDZrepro, MRI-ESM1r1, and WACCM decadal BDC regression coefficients fall within 95% confidence of MERRA, and only CCSRNIES-MIROC-3.2, LMDZrepro, and WACCM fall within 95% confidence interval of ERAI. As with the $\Delta T$ coefficient, the spread between the different decades for a single model tends to be small.

Figure 6 shows that, for all CCMs, the ensemble average decadal QBO coefficient is approximately 0 ppmv. For those CCMs that do simulate a QBO, the ensemble average coefficient is 0.02±0.03 ppmv. This is significantly smaller than the response to the QBO in the observations. Only CCSRNIES-MIROC3.2 and CMAM-CCMI decadal regressions produce QBO coefficients approaching those from both observational regressions. Again, CMAM-CCMI does not simulate a QBO, and it is not clear to us why the model does so well in this aspect of our analysis.

Previous studies found that the QBO significantly influences TTL temperatures and subsequently $[H_2O]_{entry}$ (Zhou et al., 2001; Geller et al., 2002; Liang et al., 2011), so the lack of response in the model ensemble appears to be a problem in the models. Previous studies have investigated this issue, finding that a higher vertical resolution within the stratosphere can help resolve the QBO's impact on the lower stratosphere (Rind et al., 2014; Anstey et al., 2016; Geller et al., 2016). Clearly, this needs to be investigated further.

Similar to both the trended and detrended regression analysis, we calculated the regression coefficients using standardized variables of the decadal analysis, and the values are listed in Table 4. Within most models, we see that the BDC, on decadal timescales, has the largest impact on $[H_2O]_{entry}$, with $\Delta T$ having a smaller impact.

## 5   Century and Decadal Regression Coefficient Comparison

One interesting question is whether the regression coefficients from the decadal analyses are related to regression coefficients from century regressions. To answer this, Figure 7 shows the coefficients from the trended century regressions of each CCM plotted against the median of the decadal regressions from the same CCM. Also shown is a linear least-squares fit to the points. For the $\Delta T$ coefficient, the best fit line is:

$$\beta(\Delta T, century) = 1.21 \pm 0.44 \beta(\Delta T, decade) + 0.13 \pm 0.08 \tag{2}$$

All uncertainties are 95% confidence intervals. Thus, the $\Delta T$ coefficients from the trended MLRs are slightly larger than those from the decadal MLRs. Using values of $\beta(\Delta T, decade)$ from MLS observations and this fit, we predict $\beta(\Delta T, century)$ of 0.50 $\pm$0.06 and 0.55 $\pm$0.08 ppmv K$^{-1}$ for MERRA and ERAI regressions, respectively.

For the BDC coefficient, the best fit line is:

$$\beta(BDC, century) = 1.16 \pm 0.32 \beta(BDC, decade) + 0.56 \pm 1.56 \tag{3}$$

The BDC coefficients from the trended MLRs also have a slightly larger magnitude than those from the decadal MLRs. By fitting the observed values of $\beta(BDC, decade)$ through equation 3, we predict $\beta(BDC, century)$ values of $\beta(BDC, century)$ of -3.45 $\pm$1.09 and -2.34 $\pm$1.09 ppmv (K/Day)$^{-1}$ for MERRA and ERAI regressions, respectively.

For the QBO coefficient, the best fit line is:

$$\beta(QBO, century) = 0.75 \pm 0.40 \beta(QBO, decade) + 0.004 \pm 0.01 \tag{4}$$

The QBO coefficients from the trended MLRs are slightly smaller than those from the decadal MLRs. Again, using equation 4, we predict $\beta(QBO, century)$ values of 0.09 $\pm$0.03 and 0.09 $\pm$0.02 ppmv for MERRA and ERAI regressions, respectively.

## 6   Conclusions

Climate models predict that tropical lower-stratospheric humidity ($[H_2O]_{entry}$) will increase as the climate warms, with important implications for the chemistry and climate of the atmosphere. We demonstrate in this paper that the regression used by

Dessler et al. (2013, 2014) can be used to quantify the physical processes underlying these model trends and variability. Our method is based on regressing CCM $[H_2O]_{entry}$ time series against three processes that have been shown to be important to $[H_2O]_{entry}$: tropospheric temperature ($\Delta T$), the strength of the Brewer-Dobson circulation (BDC), and the phase of the QBO. Our approach provides insight into model processes not available by simply comparing $[H_2O]entry$ to TTL temperatures.

5    We do this on two separate time-scales: 1) the $21^{st}$ century, and 2) on decadal timescales. Considering all of our analyses, we find that long-term increase in $[H_2O]_{entry}$, in the CCMs, is primarily driven by warming of the troposphere. This is partially offset in most CCMs by an increase in the strength of the Brewer-Dobson circulation, which tends to cool the tropical tropopause layer (TTL) (Randel et al., 2006a; Fueglistaler et al., 2014). For shorter-term internal variability, we find variability in the Brewer-Dobson circulation is of greater importance to the variability of $[H_2O]_{entry}$, consistent with Geller and Zhou (2007); Dessler et al. (2016). The models show little impact from the QBO.

    The coefficients from regressions of individual decades in the CCMs can be compared to coefficients from regressions of observations covering a decade. Overall, the CCM ensemble seems to reproduce $[H_2O]_{entry}$ observations well, except for the fact that the CCMs simulate little response to the QBO, in disagreement with the observations (O'Sullivan and Dunkerton, 1997; Randel et al., 1998; Dunkerton, 2001; Fueglistaler and Haynes, 2005; Choiu et al., 2006; Liang et al., 2011; Castanheira et al., 2012; Khosrawi et al., 2013; Kawatani et al., 2014; Tao et al., 2015); this appears to be a deficiency in the models.

    That said, the good agreement of the ensemble average hides some spread among the models, particularly in the response to the BDC. Of particular note, the CNRM-CM5-3 and NIWA-UKCA regressions generate positive BDC regression coefficients, contrary to the other models and contrary to our expectations. We expect a negative coefficient because it is well established that a strengthening BDC should cool the tropopause, reducing water vapor entering the stratosphere (Holton et al., 1995). The anticorrelation between BDC strength and TTL temperatures has been observed (e.g. Yulaeva et al., 1994; Flury et al., 2013), and this has been identified as the cause of the stratospheric "tape recorder" (Mote et al., 1996). This anticorrelation has also been identified as the cause of the large drop in $[H_2O]_{entry}$ around 2000 (e.g. Randel et al., 2006b; Dhomse et al., 2008).

    Our overall conclusions are encouraging — the models appear to respond to the factors that control $[H_2O]_{entry}$ in realistic ways, providing some confidence in their simulations of $[H_2O]_{entry}$. Nevertheless, our work has pointed out issues that should be resolved. Some models have clear problems, e.g., the models that predict $[H_2O]_{entry}$ will increase with a strengthening BDC. In addition, nearly the entire ensemble does not reproduce the observed variations of $[H_2O]_{entry}$ with the phase of the QBO. This analysis should help the modeling groups refine their models' simulations of the $21^{st}$ century.

## 7   Data availability

This data can be obtained through the British Atmospheric Data Center (BADC) archive.

*Author contributions.* KS and AD performed this analysis and wrote most of this manuscript. The other authors contributed information pertaining to their individual models and helped revise this paper.

*Competing interests.* The authors declare that they have no conflict of interest.

*Acknowledgements.* This work was supported by NASA grant NNX14AF15G to Texas A&M University. We acknowledge the British Atmospheric Data Center (BADC) for collecting and archiving the CCMVal and CCMI model output. We would like to thank the WACCM group at NCAR and the CNRM-CM5-3 group for model development and making their simulations available to us. Additionally, we would like to

5    thank those involved in GEOSCCM model development, the NASA MAP program, and the high-performance computing resources provided by the NASA Center for Climate Simulation (NCCS). OM acknowledges funding by the New Zealand Royal Society Marsden Fund (grant no. 12-NIW-006). OM and GZ wish to acknowledge the contribution of NeSI high-performance computing facilities to the results of this research. OM and GZ were also supported by the NZ Government's Strategic Science Investment Fund (SSIF) through the NIWA programme CACV. NZ's national facilities are provided by the NZ eScience Infrastructure and funded jointly by NeSI's collaborator institutions and

10    through the Ministry of Business, Innovation & Employment's Research Infrastructure programme (https://www.nesi.org.nz). HA acknowledges the Environment Research and Technology Development Fund, Ministry of Environment, Japan (2–1303) and NEC–SX9/A(ECO) computers at CGER, NIES. The LMDZ-REPRO contribution was supported by the European Project StratoClim (7th framework programme, Grant agreement 603557) and the Grant'SOLSPEC' from the Centre d'Etude Spatiale (CNES).

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

**Table 1.** CCMs used in this analysis. The resolution is listed as (lat x lon x number of pressure levels). 31 vertical levels indicates CCM data is given on isobaric levels, while CCMs simulating data on >31 levels are given on sigma (hybrid-pressure) levels

| CCM | Resolution | Dataset | Contains QBO | Institution | Reference(s) |
|---|---|---|---|---|---|
| CCSRNIES | 2.8° x 2.8° x 31 | CCMVal-2 | No | NIES, Tsukuba, Japan | Akiyoshi et al. (2009) |
| CCSRNIES-MIROC3.2 | 2.8° x 2.8° x 34 | CCMI-1 | Yes | NIES, Tsukuba, Japan | Imai et al. (2013); Akiyoshi et al. (2016) |
| CMAM | 5.5° x 5.6° x 31 | CCMVal-2 | No | EC, Canada | Scinocca et al. (2008) |
| CMAM-CCMI | 3.7° x 3.8° x 71 | CCMI-1 | No | EC, Canada | Jonsson et al. (2004); Scinocca et al. (2008) |
| CNRM-CM5-3 | 2.8° x 2.8° x 31 | CCMI-1 | No | Meteo-France; France | Voldire et al. (2013); Michou et al. (2011) |
| GEOSCCM | 2.0° x 2.5° x 31 | CCMVal-2 | No | NASA/GSFC, USA | Pawson et al. (2008) |
| GEOSCCM-CCMI | 2.0° x 2.5° x 72 | CCMI-1 | Yes | NASA/GSFC, USA | Molod et al. (2012, 2015); Oman et al. (2011, 2013) |
| LMDZrepro | 2.5° x 3.8° x 31 | CCMVal-2 | No | IPSL, France | Jourdain et al. (2008) |
| MRI | 2.8° x 2.8° x 31 | CCMVal-2 | Yes | MRI, Japan | Shibata and Deushi (2008) |
| MRI-ESM1r1 | 2.8° x 2.8° x 80 | CCMI-1 | Yes | MRI, Japan | Yukimoto et al. (2011, 2012); Deushi and Shibata (2011) |
| NIWA-UKCA | 2.5° x 3.8° x 31 | CCMI-1 | Yes | NIWA, NZ | Morgenstern et al. (2009, 2013) |
| WACCM | 1.9° x 2.5° x 31 | CCMVal-2 | No | NCAR, USA | Garcia et al. (2007) |

**Table 2.** Coefficients ($\beta$s) from regressions of trended $[H_2O]_{entry}$ time series, and the change in $[H_2O]_{entry}$ resulting from each process ($\beta$STD()), where STD() is the standard deviation of each trended process.

| CCM | $\Delta T$ | | BDC | | QBO | |
|---|---|---|---|---|---|---|
| | $\beta_{\Delta T}$ | $|\beta_{\Delta T}|$STD$(\Delta T)$ | $\beta_{BDC}$ | $|\beta_{BDC}|$STD(BDC) | $\beta_{\Delta QBO}$ | $|\beta_{QBO}|$STD(QBO) |
| CCSRNIES | 0.06±0.01 | 0.08±0.02 | -0.67±0.95 | 0.01±0.02 | $1.7 \times 10^{-2}$ ±0.01 | $7.9 \times 10^{-3}$ ±0.006 |
| CCSRNIES-MIROC3.2 | 0.40±0.06 | 0.39±0.06 | -3.4±1.9 | 0.11±0.06 | $3.5 \times 10^{-2}$ ±0.04 | $2.2 \times 10^{-2}$ ±0.02 |
| CMAM | 0.26±0.02 | 0.39±0.03 | -5.7±1.1 | 0.07±0.01 | $8.0 \times 10^{-4}$ ±0.03 | $4.7 \times 10^{-4}$ ±0.02 |
| CMAM-CCMI | 0.22±0.05 | 0.21±0.05 | -3.8±2.6 | 0.06±0.04 | $8.2 \times 10^{-2}$ ±0.04 | $3.8 \times 10^{-2}$ ±0.02 |
| CNRM-CM5-3 | 0.27±0.13 | 0.26±0.13 | 3.7±5.4 | 0.09±0.13 | $1.9 \times 10^{-2}$ ±0.07 | $4.9 \times 10^{-3}$ ±0.02 |
| GEOSCCM | 0.38±0.03 | 0.37±0.03 | -6.7±0.82 | 0.21±0.03 | $-1.3 \times 10^{-2}$ ±0.01 | $3.2 \times 10^{-3}$ ±0.003 |
| GEOSCCM-CCMI | 0.27±0.03 | 0.27±0.02 | -6.6±0.96 | 0.17±0.03 | $5.2 \times 10^{-3}$ ±0.02 | $2.8 \times 10^{-3}$ ±0.01 |
| LMDZrepro | 0.55±0.04 | 0.72±0.05 | -8.3±2.1 | 0.10±0.04 | $1.4 \times 10^{-2}$ ±0.04 | $6.8 \times 10^{-3}$ ±0.02 |
| MRI | 0.57±0.03 | 0.58±0.03 | -12.±1.3 | 0.34±0.04 | $-4.1 \times 10^{-3}$ ±0.03 | $2.0 \times 10^{-3}$ ±0.01 |
| MRI-ESM1r1 | 0.36±0.05 | 0.36±0.05 | -3.1±1.4 | 0.12±0.05 | $1.7 \times 10^{-2}$ ±0.03 | $9.5 \times 10^{-3}$ ±0.02 |
| NIWA-UKCA | 0.20±0.07 | 0.20±0.07 | 4.3±4.6 | 0.06±0.07 | $-1.0 \times 10^{-2}$ ±0.07 | $5.9 \times 10^{-3}$ ±0.04 |
| WACCM | 0.24±0.04 | 0.21±0.03 | -3.5±1.2 | 0.05±0.02 | $1.5 \times 10^{-2}$ ±0.03 | $4.7 \times 10^{-3}$ ±0.008 |

The units of $\Delta T$, BDC, and QBO are ppmv K$^{-1}$, ppmv (K/Day)$^{-1}$, and ppmv, while the units of $\beta_{\Delta T}$STD$(\Delta T)$, $\beta_{BDC}$STD(BDC), and $\beta_{QBO}$STD(QBO) are all ppmv. The uncertainty is the 95% confidence interval.

**Table 3.** Coefficients ($\beta$s) from regressions of detrended $[H_2O]_{entry}$ time series, and the change in $[H_2O]_{entry}$ resulting from each process ($\beta$STD()), where STD() is the standard deviation of each detrended process.

| | Detrended Regression | | | | | |
|---|---|---|---|---|---|---|
| CCM | $\Delta T$ | | BDC | | QBO | |
| | $\beta_{\Delta T}$ | $|\beta_{\Delta T}|$STD($\Delta T$) | $\beta_{BDC}$ | $|\beta_{BDC}|$STD(BDC) | $\beta_{\Delta QBO}$ | $|\beta_{QBO}|$STD(QBO) |
| CCSRNIES | 0.05±0.02 | 0.02±0.006 | -0.67±0.67 | $7.1\text{x}10^{-3}$ ±0.005 | $1.7\text{x}10^{-2}$ ±0.01 | $3.6\text{x}10^{-3}$ ±0.003 |
| CCSRNIES-MIROC3.2 | 0.30±0.05 | 0.08±0.01 | -4.3±0.83 | 0.08±0.02 | $2.8\text{x}10^{-2}$ ±0.01 | $1.7\text{x}10^{-2}$ ±0.009 |
| CMAM | 0.26±0.03 | 0.10±0.01 | -5.3±0.84 | 0.05±0.008 | $7.0\text{x}10^{-4}$ ±0.02 | $1.9\text{x}10^{-4}$ ±0.006 |
| CMAM-CCMI | 0.26±0.05 | 0.05±0.01 | -3.7±1.1 | 0.04±0.01 | $7.7\text{x}10^{-2}$ ±0.04 | $2.9\text{x}10^{-2}$ ±0.005 |
| CNRM-CM5-3 | 0.19±0.05 | 0.08±0.01 | 0.20±1.1 | $2.5\text{x}10^{-3}$ ±0.01 | $-3.3\text{x}10^{-2}$ ±0.01 | $7.1\text{x}10^{-3}$ ±0.003 |
| GEOSCCM | 0.31±0.04 | 0.08±0.009 | -6.6±0.65 | 0.09±0.009 | $-1.0\text{x}10^{-2}$ ±0.01 | $1.9\text{x}10^{-3}$ ±0.002 |
| GEOSCCM-CCMI | 0.25±0.04 | 0.07±0.01 | -7.1±0.71 | 0.17±0.03 | $4.4\text{x}10^{-3}$ ±0.01 | $2.3\text{x}10^{-3}$ ±0.007 |
| LMDZrepro | 0.59±0.05 | 0.25±0.02 | -5.4±1.1 | 0.05±0.02 | $-5.5\text{x}10^{-3}$ ±0.03 | $2.3\text{x}10^{-3}$ ±0.01 |
| MRI | 0.52±0.03 | 0.18±0.02 | -11.±1.0 | 0.24±0.02 | $-4.6\text{x}10^{-4}$ ±0.02 | $2.2\text{x}10^{-4}$ ±0.01 |
| MRI-ESM1r1 | 0.33±0.05 | 0.09±0.01 | -4.3±0.61 | 0.10±0.01 | $5.5\text{x}10^{-3}$ ±0.01 | $3.0\text{x}10^{-3}$ ±0.007 |
| NIWA-UKCA | 0.15±0.08 | 0.04±0.02 | 2.9±1.6 | 0.04±0.02 | $-1.0\text{x}10^{-2}$ ±0.02 | $5.9\text{x}10^{-3}$ ±0.01 |
| WACCM | 0.23±0.05 | 0.06±0.01 | -3.5±0.80 | 0.04±0.01 | $1.5\text{x}10^{-2}$ ±0.02 | $2.8\text{x}10^{-3}$ ±0.004 |

The units of $\Delta T$, BDC, and QBO are ppmv K$^{-1}$, ppmv (K/Day)$^{-1}$, and ppmv, while the units of $\beta_{\Delta T}$STD($\Delta T$), $\beta_{BDC}$STD(BDC), and $\beta_{QBO}$STD(QBO) are all ppmv. The uncertainty is the 95% confidence interval.

**Table 4.** Median coefficients from the decadal regressions of $[H_2O]_{entry}$ monthly anomalies, and the change in $[H_2O]_{entry}$ resulting from each process ($\beta$STD()), where STD() is the standard deviation of each decadal process.

| Decadal Regressions | | | | | | |
|---|---|---|---|---|---|---|
| CCM | $\Delta T$ | | BDC | | QBO | |
| | $\beta_{\Delta T}$ | $|\beta_{\Delta T}|$STD($\Delta T$) | $\beta_{BDC}$ | $|\beta_{BDC}|$STD(BDC) | $\beta_{\Delta QBO}$ | $|\beta_{QBO}|$STD(QBO) |
| CCSRNIES | 0.03±0.04 | 8.7x10$^{-3}$ ±0.01 | -1.23±1.34 | 0.01±0.02 | 5.26x10$^{-3}$ ±0.02 | 1.5x10$^{-3}$ ±0.005 |
| CCSRNIES-MIROC3.2 | 0.10±0.17 | 0.03±0.02 | -3.29±1.44 | 0.10±0.04 | 6.05x10$^{-2}$ ±0.01 | 5.7x10$^{-2}$ ±0.02 |
| CMAM | 0.19±0.09 | 0.05±0.03 | -6.06±1.34 | 0.07±0.02 | 2.75x10$^{-3}$ ±0.03 | 9.4x10$^{-4}$ ±0.004 |
| CMAM-CCMI | 0.01±0.10 | 3.5x10$^{-3}$ ±0.02 | -4.70±1.29 | 0.07±0.03 | 6.13x10$^{-2}$ ±0.01 | 3.0x10$^{-2}$ ±0.02 |
| CNRM-CM5-3 | 0.06±0.14 | 0.01±0.03 | 2.89±1.44 | 0.05±0.02 | 1.84x10$^{-2}$ ±0.02 | 4.9x10$^{-3}$ ±0.01 |
| GEOSCCM | 0.17±0.10 | 0.04±0.02 | -6.31±1.19 | 0.13±0.03 | -1.47x10$^{-2}$ ±0.03 | 4.9x10$^{-3}$ ±0.005 |
| GEOSCCM-CCMI | 0.11±0.16 | 0.02±0.03 | -8.00±1.89 | 0.18±0.06 | 2.42x10$^{-2}$ ±0.02 | 1.8x10$^{-2}$ ±0.01 |
| LMDZrepro | 0.31±0.19 | 0.11±0.08 | -2.71±2.71 | 0.07±0.05 | 1.27x10$^{-2}$ ±0.01 | 6.9x10$^{-3}$ ±0.03 |
| MRI | 0.35±0.09 | 0.12±0.04 | -8.78±2.91 | 0.25±0.07 | -6.56x10$^{-3}$ ±0.06 | 4.6x10$^{-3}$ ±0.03 |
| MRI-ESM1r1 | 0.19±0.04 | 0.05±0.01 | -4.72±0.71 | 0.13±0.03 | 1.17x10$^{-2}$ ±0.03 | 8.9x10$^{-3}$ ±0.02 |
| NIWA-UKCA | 0.05±0.29 | 0.01±0.06 | 2.11±3.26 | 0.04±0.05 | -1.88x10$^{-2}$ ±0.04 | 1.5x10$^{-2}$ ±0.03 |
| WACCM | 0.15±0.12 | 0.03±0.03 | -2.25±0.85 | 0.05±0.02 | 3.84x10$^{-2}$ ±0.03 | 9.1x10$^{-3}$ ±0.007 |
| MLS/ERAI | 0.34±0.17 | 0.11±0.05 | -2.5±0.83 | 0.17±0.06 | 1.1x10$^{-1}$ ±0.04 | 0.11±0.05 |
| MLS/MERRA | 0.30±0.20 | 0.11±0.07 | -3.5±1.6 | 0.15±0.07 | 1.2x10$^{-1}$ ±0.05 | 0.12±0.06 |

The units of $\Delta T$, BDC, and QBO are ppmv K$^{-1}$, ppmv (K/Day)$^{-1}$, and ppmv, while the units of $\beta_{\Delta T}$STD($\Delta T$), $\beta_{BDC}$STD(BDC), and $\beta_{QBO}$STD(QBO) are all ppmv. The uncertainty represents the variability (one standard deviation) in the set of coefficients produced by each CCM. For observations, the error bars represent 95% confidence.

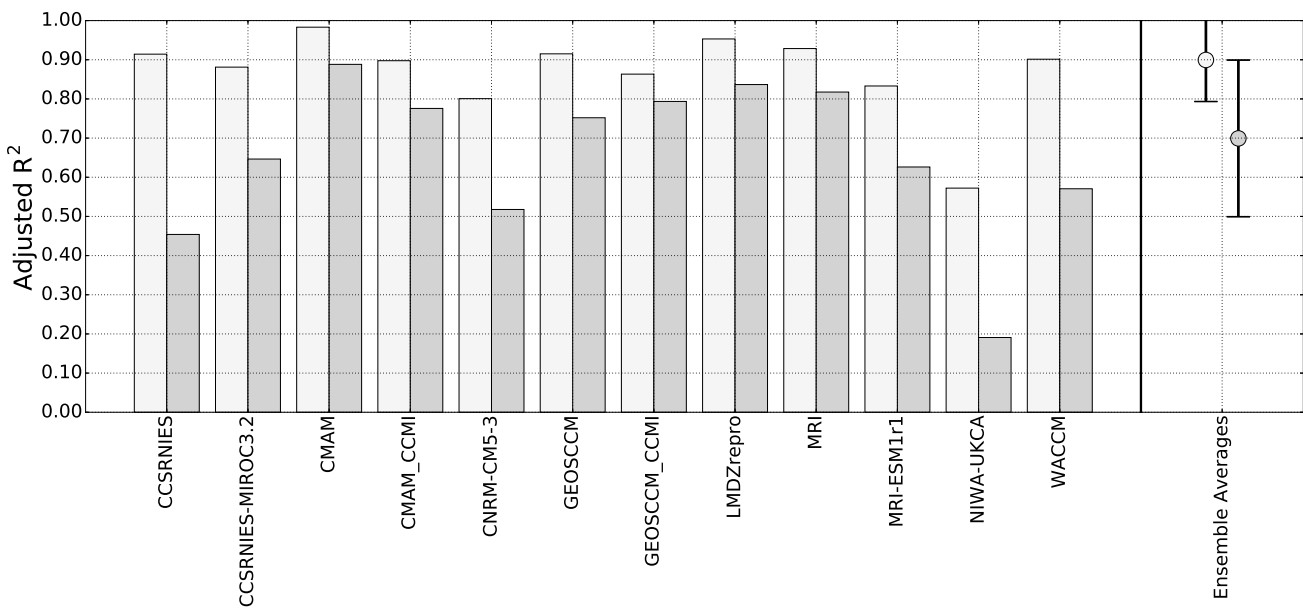

**Figure 1.** Bars corresponds to trended (light grey) and detrended (dark grey) adjusted $R^2$ values for annual-averaged data. The circles represent the ensemble mean, with error bars indicating $\pm$ one standard deviation of the CCM ensemble.

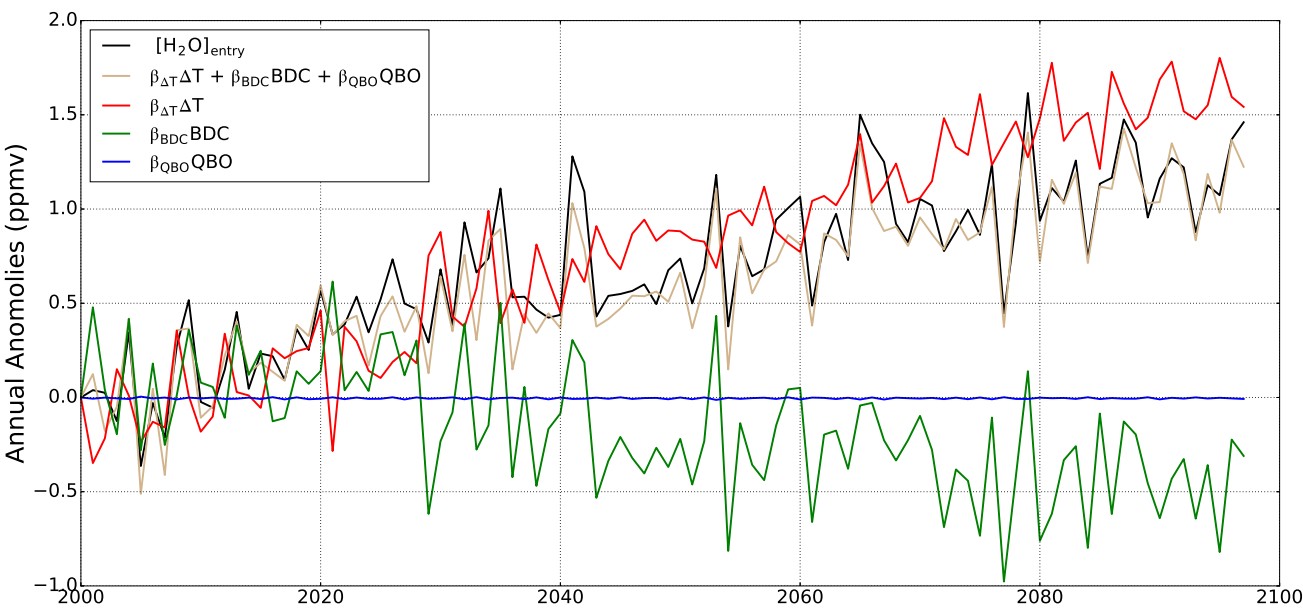

**Figure 2.** Time series of annual-averaged anomalies of $[H_2O]_{entry}$ from the MRI (black), and its reconstruction using a multivariate linear regression (brown). The red, green, and blue lines are the $\Delta T$, BDC, and QBO terms from the regression, respectively.

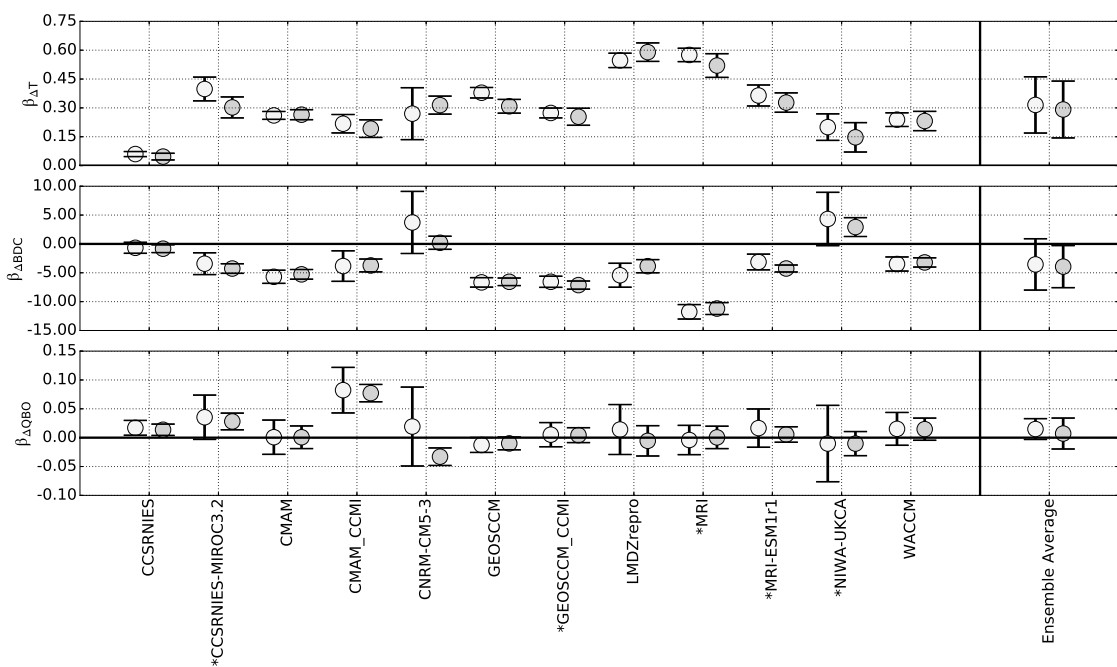

**Figure 3.** Circles show detrended (light grey) and trended (dark grey) coefficients for each model; error bars correspond to $95^{th}$ percentile confidence interval bounding each regression coefficient. An asterisk indicates models simulating a QBO. The ensemble mean corresponds to the average of all model coefficients. The ensemble mean coefficients are also represented by a circle, with associated error bars correspond to $\pm$one standard deviation of the ensemble. The units of $\beta_{\Delta t}$, $\beta_{BDC}$, and $\beta_{QBO}$ are ppmv/K, ppmv/(K/Day), and ppmv, respectively.

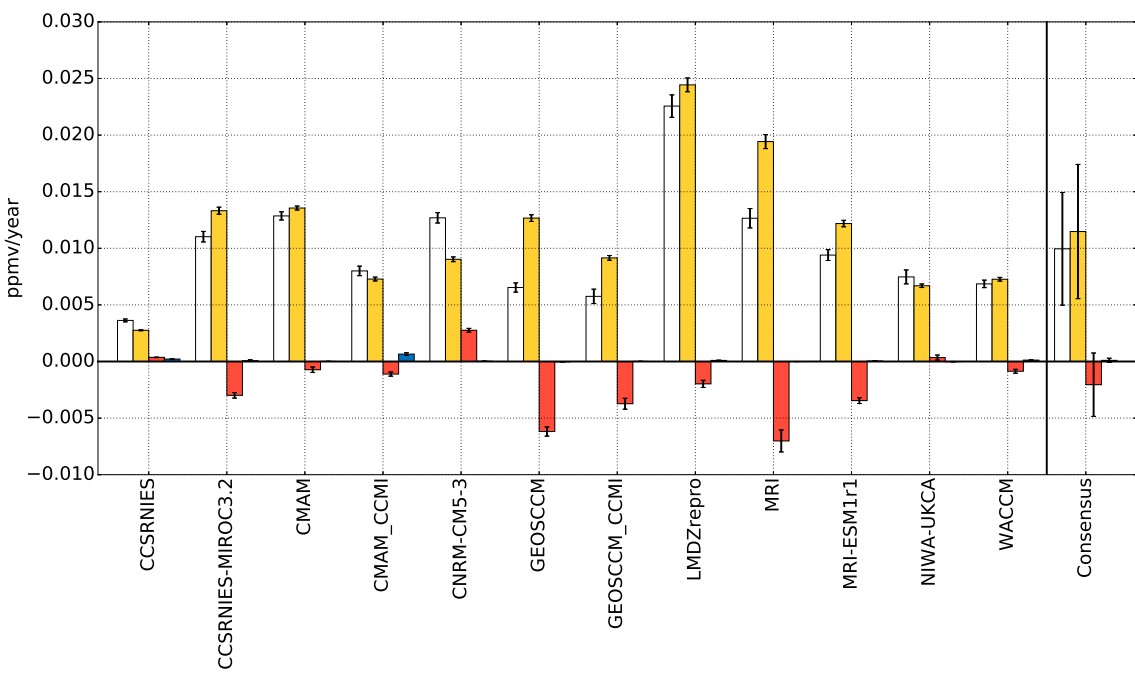

**Figure 4.** Trends in $[H_2O]entry$ (white) resulting from $\Delta T$ (yellow), BDC (red), and QBO (blue) predictor time series assuming the other predictors are held constant. Error bars represent 95% uncertainty. For many models, the contribution of the QBO is too small to be seen.

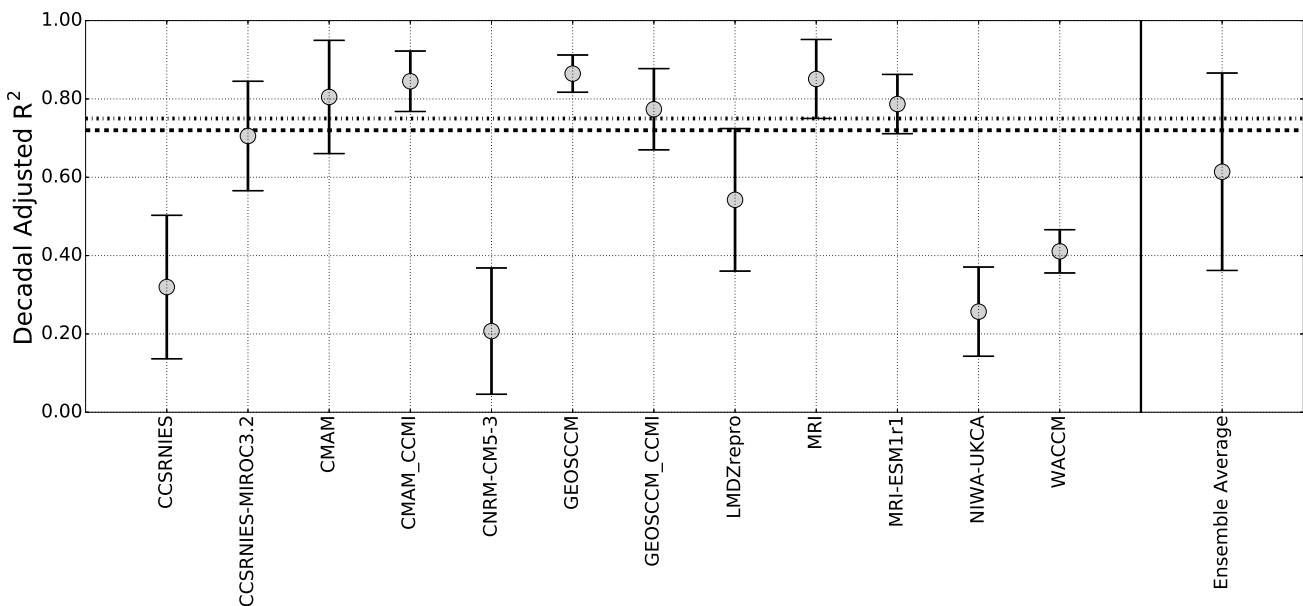

**Figure 5.** Circles represent the median of the adjusted $R^2$ value of the decadal fits. Errors correspond to the $\pm$ one standard deviation of the adjusted $R^2$ values. The CCM ensemble average is also plotted, along with error bars corresponding to $\pm$ one standard deviation of ensemble set of decadal adjusted $R^2$ values. The lines are adjusted $R^2$ values from observations combined with reanalysis (ERAI (dotted) and MERRA (dashed)) from Dessler et al. (2014).

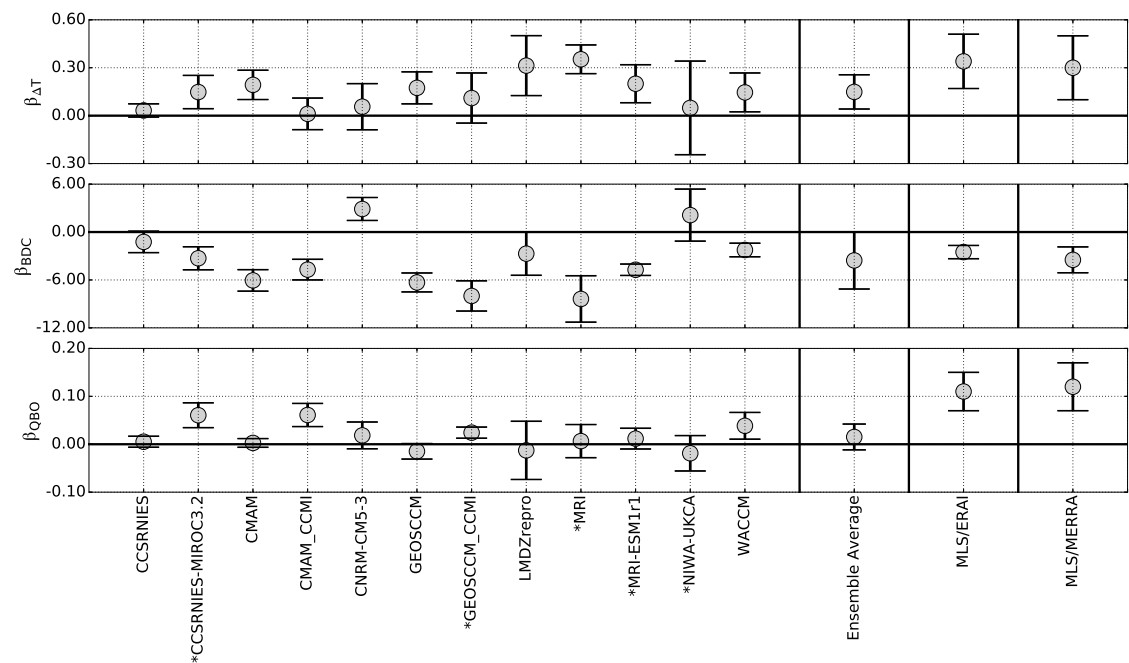

**Figure 6.** Circles represent the median decadal regression coefficient from each CCM, and error bars correspond to $\pm$ one standard deviation. An asterisk indicates that the model simulates a QBO. The ensemble mean corresponds to an average of all model coefficients. The ensemble mean coefficients are also represented by a circle, with associated error bars correspond to $\pm$one standard deviation of the ensemble set of coefficients. Estimates from observations combined with reanalysis (Dessler et al., 2014) shown, along with $95^{th}$ percentile confidence interval. The units of $\beta_{\Delta t}$, $\beta_{BDC}$, and $\beta_{QBO}$ are ppmv/K, ppmv/(K/Day), and ppmv, respectively.

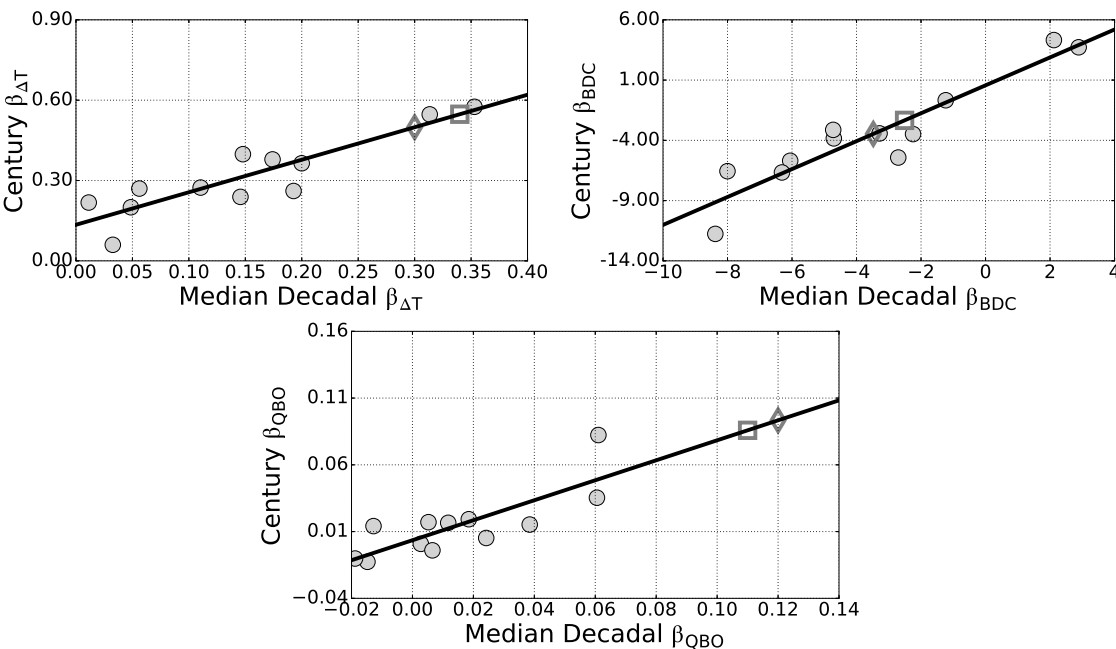

**Figure 7.** (Top Left) Scatter plots of trended $\Delta T$ regression coefficients (ppmv K$^{-1}$) vs. median decadal $\Delta T$ regression coefficients (ppmv K$^{-1}$) from each CCM. (Top Right) Same as top, but for BDC coefficients. (Bottom Middle) Same as top left and top right, but for QBO coefficient . Black lines in all plots correspond to a best fit line between the trended and decadal coefficients, and the observational coefficients ERAI (square) and MERRA (diamond) are fitted to each line (from Dessler et al. (2014)).