# Peer review of "Testing chemistry-climate models' regulation of tropical lower-stratospheric water vapor"

_Atmospheric Chemistry and Physics, 2016_

## Referee Comment (RC1) · Kevin M. Smalley et al. · 7 Dec 2016

This study deals with the effect of climate change on stratospheric water vapor, in particular the effect that changes in e.g. the tropospheric temperature or the Brewer-Dobson circulation have on lower stratospheric water vapor. In my opinion, this is a relevant topic and the study is worth publishing.

I found the paper readable and well-structured and most of the used methods are scientifically sound. Unfortunately, I think there is a misconception in transferring your results from the trended to the detrended regression analysis, which affects your main conclusions (see major comment 1). In addition, there is some lack of discussion and interpretation of your results, in particular at places where I think it is needed to support your main conclusions. Unfortunately, that means that in my opinion you currently give

not enough information in the paper to support your two main conclusions (A. "The long-term trend in stratospheric humidity is driven by the warming of the troposphere", B. "A stratospheric water vapor feedback exists, where a warming climate increases stratospheric water vapor, leading to further tropospheric warming.")

I think some more discussion and information and a correct treatment of the detrended analysis would solve this issue and make it easier for the reader to assess your results. Thus, I suggest to publish the paper after the necessary revisions, which are probably relatively easy to implement.

**Major comments**

- 1. There is a misconception in transferring your results from the trended to the detrended regression analysis. The problem here is, that you use the regression coefficients $\beta$ directly in some places to analyse your results (e.g. in Table 2, 3, 4 and in Fig. 3 and 5 and accompanying text) and not either the explained variance (i.e. VAR($\beta_{\Delta T}\Delta T$)) or the regression coefficient multiplied by the standard deviation of the explanatory variable (i.e. $\beta_{\Delta T}$STD($\Delta T$)). At some places (Page 4, line 5–6) you look at these quantities, but unfortunately at Page 4, line 14–15, you draw the conclusion "This confirms the stratospheric water vapor feedback [...]" from the similarity of the *regression coefficients* in the trended and the detrended analysis.

  Unfortunately, this is an invalid conclusion. Even if the regression coefficients would stay exactly identical, the percentage of explained variance that an explanatory time series explains of the total explained variance $R^2$ can change dramatically between the trended and detrended regression analysis. An obvious example is an explanatory time series with a large trend and a small interannual variability. An explanatory timeseries like this will likely contribute a large explained variance to the trended regression analysis, but a small explained variance (in percent) to a detrended regression analysis, while its regression coefficient may be very similar in the trended and detrended analysis. Unfortunately, your example time series for $\Delta T$ in Fig. 2 looks a little like this (compared to the variance and trend of the BDC time series). Since we agree that you can't really use the trended analysis to confirm your main conclusion (Page 3, line 26–27), you have to base your conclusion that changes in tropospheric temperature cause changes in stratospheric water vapor on the detrended regression analysis. That means you have to confirm that a large part of the interannual variability of stratospheric water vapor in the detrended regression analysis comes from interannual varibaility in the $\Delta T$ term. That still will not be a proof of causality, but will put much more confidence in your main conclusions.

Additional remark 1: Since it is known that variability in stratospheric water vapor comes from variations in the tropopause temperature (more exactly: Lagrangian dry points, see e.g. Fueglistaler, 2013), it would put much more confidence in your main conclusions if you show that tropospheric temperature and tropopause temperatures correlate in your models.

Additional remark 2: Giving values as explained variances makes it easier to compare values between different time series as $\Delta T$ and BDC. In the moment, it is easy to compare between models in the rows of your tables, but impossible to do that between the columns of your tables.

Additional remark 3: I have to emphasize that I am pretty sure that the trend in $\Delta T$ and in stratospheric water vapor are causally connected, I just think that a trended regression analysis is not the tool to show that. You have to avoid the impression that your trended regression analysis is a proof of that.

My suggestion is the following: Add values for the explained variance of the explanatory time series (e.g. in ppm$^2$) to the tables 2, 3 and 4 (you can keep the regression coefficients or replace them by these values). Alternatively, you can add values for the regression coefficients multiplied by the standard deviations of the explanatory time series to the tables. Both explained variance and standard

deviation have advantages and disadvantages: The explained variances of the explanatory timeseries add up to the overall explained variance (under the assumption that the explanatory timeseries are uncorrelated), but values in ppm$^2$ are not very intuitive. Standard deviations are more intuitive, but don't add up. Thus, I will not give a recommendation what is better here. Next, change Figure 3 to show explained variances or standard deviations, or add an additional figure doing this. Then, base your discussion on the explained variances, where it does matter for your conclusions (e.g. in section 3.2).

- 2. It is not straightforward that more stratospheric water vapor means more warming of the troposphere, and there is not enough discussion in your paper in the moment to support your main conclusion "A stratospheric water vapor feedback exists, where a warming climate increases stratospheric water vapor, leading to further tropospheric warming". Please at least discuss the literature on that shortly (e.g. Oinas et al., 2001, Solomon et al., 2010). That would give much more confidence that this statement is actually correct. Is the feedback by an increase in downward longwave radiation from the stratosphere? That does not seem to be straightforward to me. One the one hand, you have more water vapor to emit radiation. On the other hand, the stratosphere gets cooler, which reduces radiation. In a simple picture, where water vapor only emits longwave radiation and the stratosphere is heated by shortwave radiation by ozone, wouldn't the outgoing longwave radiation from a layer where you add more water vapor just stay constant to mantain radiative equilibrium, by a lowered radiative equilibrium temperature?

- 3. It seems to me that you take the positive correlation between tropospheric temperature and stratospheric water vapor as very obvious. However, this is not simple and obvious at all. Again, discuss the literature on that shortly, and try to avoid the impression that this is an obvious fact.

Certainly, there is the simple way of thinking about it as shown in Fig. 6 of Shepherd (2002). But the radiative balance at the tropical tropopause is rather delicate, since we are near the line of zero radiative heating and it is difficult to say a priori (without detailed radiative transfer calculations) what will happen when greenhouse gases increase. Different greenhouse gases have rather different behaviour near the tropopause, e.g. more water vapor will cool in the longwave region, while more $CO_2$ will heat in the shortwave region, and heat or cool in the longwave region dependent on altitude, e.g. Gettelman et al., 2004. And I wonder what the effect of convection is on that (which has its main detrainment level far below the tropopause but reaches up to the tropopause). Papers that discuss how complicated the influences on tropopause temperature are, are e.g. Lin et al. (2016) or Gettelman et al. (2004).

While there are relatively clear positive trends in tropopause temperature and tropopause altitude in most models as a response to increase in greenhouse gases (e.g. Kim et al., 2013, Gettelman et al., 2010) (probably caused by increased heating by $CO_2$), observed trends in these quantities are more inconclusive (see e.g. Fueglistaler et al., 2013) and it is even difficult to prove unequivocally that the observed water vapor can be solely explained by the observed tropopause temperatures and location of the LDPs.

- 4. Since multiple regression can only show correlation but not causality, some more discussion on the supposed reasons for the correlations would be very helpful, in particular for the $\Delta T$ term. In my opinion, it should also be discussed that the reasons for a correlation can be very different in a model and in reality (i.e. based on observations). Just to give a simple example: The correlation between tropospheric temperatures and stratospheric water vapor can possibly be caused by excessive transport or diffusion of water vapor over the tropopause in the models (see e.g. Hardiman et al., 2015 for a paper discussing this): Higher tropospheric temperatures means more moisture, which then could be transported by spurious vertical numerical diffusion into the stratosphere. A way to test for

things like this could be e.g. to look at the tropical tropopause temperatures and their correlation to tropospheric temperatures and stratospheric water vapor.

- 5. Relating to this: There is a lack of information on the model performance and parameterizations of the used models. At least some information of the following list would be very helpful to assess your results. I acknowledge that it would be a lot of work to answer all of these questions for all of the models. But I think that there should be at least some discussion about how the processes in the model can affect the results. Of course, I don't want you to discuss all of these issues in detail, but to discuss things that are important for your results, i.e. take the list below as a list of suggestions.

    - 5a. What is the tropopause temperature in the models, and how does it compare to measurements in terms of bias, annual cycle and trends? Can it explain the water vapor in the model or are there additional processes at work?
    - 5b. How well is the Brewer-Dobson circulation represented?
    - 5c. How is convection parameterized? How well does it compare to observations? Is there overshooting?
    - 5d. How is radiation parameterized? What is the effect of clouds on radiation?
    - 5e. What is the spatial pattern of Local dry points (LDPs) in the models and compared to reality? Can a shift in their distribution cause the correlation?
    - 5f. Effect of (spurious) diffusion and transport

**Specific comments**

- Page 1, line 1 and page 2, line 14: Please give a citation here, e.g. Gettelman et al. (2010) (e.g. Fig. 17) or Kim et al. (2013).

- Page 1, line 4: You probably mean stratospheric humidity. Please clarify.

- Same sentence: In case you base that statement on your trended regression analysis, is it really correct? Correlation does not imply causality, especially in a trended regression analysis. The statement that you give on page 3, line 26–27 is a direct contradiction of what you state here. In fact, I think you cannot support that statement with the information you currently give in the paper. I would try to phrase that more carefully, e.g. by speaking of correlations, or make clear that this conclusion comes not from your trended regression analysis, but from some other source.

- Page 1, line 3–5: Since there is the counteracting trend from increased cooling by the BDC (as you note here and is seen in your Figure 2), can you really make the statement that the net trend in humidity is primarily driven by tropospheric warming (that would imply to me that, say, something like 80 % or 90 % of the net trend come from the $\Delta T$ term)? It seems to me that the trend by the BDC is in the same order of magnitude (but that the net effect of both trends is normally positive). Please add a figure showing the trends by the BDC term and the $\Delta T$ term for every model to quantify the trends and to underpin your statement. I think such a figure is probably easy to add.

- Page 1, line 6–7: I don't quite understand why you split your timeseries into 10 year chunks? Would it not be ok to compare the 100 year timeseries to the 10 years of observations directly?

- Page 1, line 8: It is not clear to me what exactly you are referring to. Is it really that new to apply a linear regression model to these data (one of your own papers did that already: Dessler et al., 2013)? I either suggest to delete the last sentence of the abstract or to be more specific here: What is superior to what?

- Page 1, line 11 to Page 2, Line 2: Instead of speaking of the TTL temperatures

as the determining factor, one can get more specific here. It is the temperature of the coldest point along each air mass trajectory (i.e. the Lagrangian dry point) which determines the stratospheric water vapor (except for direct injection by overshooting). In many cases this temperature will be reached at or near the tropical tropopause.

- Page 1, line 18–19: Would be nice to add a citation here, e.g. one of the Fueglistaler papers.

- Page 2, line 1–2: No, it doesn't imply that. See general comment 2. In addition: the local effect of more water vapor is more cooling in the stratosphere, so it is better to be more specific and to write "further tropospheric warming".

- Page 3, line 14: Probably it is better to speak of "autocorrelation in the residuals" than of "autocorrelation of the timeseries", since it is only the remaining autocorrelation in the residuals that affects the uncertainties.

- Page 3, line 18–19: You are aware that subtracting a constant does only change $\beta_0$, but does not change anything else in the regression analysis?

- Page 4, line 14–15: No it doesn't confirm that, see major comment 1.

- Page 4, line 18–19: This doesn't really tell you anything, see Page 4, line 14–15.

- Section 4: I don't really get the additional benefit of splitting the timeseries into 10 year chunks. Wouldn't a direct comparison of the observational 10 year time series and the model 100 year time series give all information that is important?

- Page 7, line 8–9: I find the statement that you can assess the realism of the model trend by a linear regression somewhat problematic. If there is a trend in stratospheric water vapor in the models, and there is a trend in one of the explanatory variables, the explanatory variable will try to fit this trend, whatever

the magnitude is and whatever the underlying physical reason of the trend is. If it turns out then, that the fit of the interannual variability is also good, that may give you some confidence. But in general, you have always the problem that a linear regression analysis does not tell you anything about causal relationships.

Page 7, line 13: See 2nd specific comment for Page 1, line 4. I would phrase that more carefully.

Page 7, 13–16: I think it would make sense to cite some studies here and to discuss your results in comparison to other studies (briefly), e.g. studies that deal with the absence of the QBO in many models, that show the influence of the BDC on tropopause temperatures and its increasing trend etc.

Page 7, line 21: I would agree, but I would base that statement mainly on the detrended regression analysis. If there is a good overall fit of the detrended model, you can have some confidence that the explanatory timeseries actually are relevant processes for the explained variable, and that the magnitude of their fit does tell you something. Since regression analysis does not tell you anything about causal relationships however, you need to put some a priori knowledge into that. For that reason, I would be very careful to interpret the trended analysis, since there is the danger that there is no causal relationship between the trends (and the trends lead to a correlation between explanatory variables, which can make the magnitude of the fit for these variables a little bit arbitrary in the worst case).

Page 7, line 22: That is a conclusion I would mainly draw from comparison with observations or testing the model's processes. A regression model can only help you in confirming this. E.g. What would happen if all models would overestimate variability of water vapor in the future? Your fit coefficients would get larger to try fit this variability better. Do you learn from that that the model does a good job?

Page 7, line 22–23: This might however also be a deficiency of the regression approach, e.g. an explanatory variable that is no perfect proxy for the BDC, or

that the trends dominate the fit (which gives rise to correlation between the explanatory variables), leading to uncertainties in the magnitude of the fit for the BDC.

**Technical corrections**

- In the title you write "lower-stratospheric", later you write "lower stratospheric". Would be nice to have that consistent.

- Page 2, line 27: Change "ozone depleting substance" to "ozone depleting substances".

- Page 2, line 31: Change "described described" to "described".

- Page 5, line 12: A period is missing ("...regression. However...").

- Page 7, line 22: Change "appear do" to "appear to do"

**References**

- Fueglistaler et al., J. Geophys. Res., 118, doi:10.1002/jgrd.50157, 2013

- Gettelman et al., J. Geophys. Res., 109, doi:10.1029/2003JD004190, 2004

- Gettelman et al., J. Geophys. Res., 115, doi:10.1029/2009JD013638, 2010

- Hardiman et al., J. Clim., 28, 6516, 2015

- Kim et al., J. Geophys. Res., 118, doi:10.1002/jgrd.50649, 2013

- Lin et al., J. Clim., doi:10.1175/JCLI-D-16-0457.1, 2016

- Oinas et al., Geophys. Res. Lett., doi:10.1029/2001GL013137, 2001
- Shepherd, J. Met. Soc. Japan, 80, 4B, 769, 2002

- Solomon et al., Science, 327, 1219, 2010

---

## Referee Comment (RC2) · Anonymous Referee #2 · 15 Jan 2017

Smalley et al. analyse CCM model predictions of stratospheric water changes over the 21st century. A multivariate linear regression is applied to the models' stratospheric water entry mixing ratios ("[H2O]entry"), with the explanatory variables being a "tropospheric temperature index", a "Brewer Dobson strength" index, and a QBO index; this analysis follows the method of Dessler et al. (2013). Overall, the analysis is straightforward, and the results are clearly described. I do not comment on the aspects of the statistical analysis brought up by the other reviewer.

However, this reviewer cannot quite see that "Our approach provides more insight into model processes than simply comparing [H2O]entry or TTL temperatures." (Page 7/Line 19). Rather, the paper is somewhat superficial (it certainly does not help that

(Page 2/Line 13): "Finally, a warmer troposphere tends to increase [H2O]entry, although whether this is through influence on TTL temperatures or some other mechanism such as convective ice lofting, is not clear."), and results are few. It would be great if the authors would work out the connection between tropopause temperatures and [H2O]entry in the models, and the connection between "tropospheric temperature" and tropopause temperature.

The QBO results would also deserve some further analysis - for the 21st century analysis, annual mean data is analysed. This evidently removes much of the variance associated with the QBO, and it appears that the lack of influence of the QBO (as e.g. shown in Figure 2) is due to a lack of a trend in the QBO index. This evidently begs the question why the model does not have a QBO trend when it has been argued that the tropospheric expansion associated with global warming would have an impact on the lower stratospheric QBO - and as such would be reflected in the QBO index. While this may not have an impact on [H2O]entry (because the QBO influence at the rising tropopause level main remain constant over time), it would be useful to have some more information why the QBO index (as e.g. shown in Figure 2) does not show a trend. Two additional minor comments: Please provide a reference for the statement that "Virtually all climate models ..." (page 2/Line 14); and some more information about the differences in results for models that participated in CCMI-I and CCMVal-2 would be useful.

---

## Author Comment (AC1) · 20 Mar 2017

**acp reviewer1 revisions**

Kevin Smalley

March 2017

We thank Reviewer 1 for their thorough review. In our response, the reviewer's comments are bolded, our answers are normal weight, and anything that we change in the paper is italicized.

**Major Comment 1: There is a misconception in transferring your results from the trended to the detrended regression analysis. The problem here is, that you use the regression coefficients $\beta$ directly in some places to analyse your results (e.g. in Table 2, 3, 4 and in Fig. 3 and 5 and accompanying text) and not either the explained variance (i.e. $\mathbf{VAR}(\beta\Delta T \ \Delta T)$) or the regression coefficient multiplied by the standard deviation of the explanatory variable (i.e. $\beta\Delta T \ \mathbf{STD}(\Delta T)$). At some places (Page 4, line 5–6) you look at these quantities, but unfortunately at Page 4, line 14–15, you draw the conclusion "This confirms the stratospheric water vapor feedback [. . . ]" from the similarity of the regression coefficients in the trended and the detrended analysis. Unfortunately, this is an invalid conclusion. Even if the regression coefficients would stay exactly identical, the percentage of explained variance that an explanatory time series explains of the total explained variance $\mathbf{R}^2$ can change dramatically between the trended and detrended regression analysis. An obvious example is an explanatory time series with a large trend and a small interannual variability. An explanatory time series like this will likely contribute a large explained variance to the trended regression analysis, but a small explained variance (in percent) to a detrended regression analysis, while its regression coefficient may be very similar in the trended and detrended analysis. Unfortunately, your example time series for $\Delta T$ in Fig. 2 looks a little like this (compared to the variance and trend of the BDC time series). Since we agree that you can't really use the trended analysis to confirm your main conclusion (Page 3, line 26– 27), you have to base your conclusion that changes in tropospheric temperature cause changes in stratospheric water vapor on the detrended regression analysis. That means you have to confirm that a large part of the interannual variability of stratospheric water vapor in the detrended regression analysis comes from interannual variability in the $\Delta T$ term. That still will not be a proof of causality,**

**but will put much more confidence in your main conclusions**

The reviewer makes an excellent point here. In response, we have added new columns to Tables 2, 3, and 4 that show the correlation coefficient scaled by the standard deviation of the predictor time series. This is described on page 5, lines 1-3 of the manuscript, and we have modified our discussion to incorporate these values (page 5, lines 3-10 for the century regressions, and page 6, lines 32-35 for the decadal regressions). Additionally, we added a sentence to the conclusions, page 7, lines 29-30, to summarize the results.

**Additional remark 1: Since it is known that variability in stratospheric water vapor comes from variations in the tropopause temperature (more exactly: Langrarian dry points, see e.g. Fueglistaler, 2013), it would put much more confidence in your main conclusions if you show that tropospheric temperature and tropopause temperatures correlate in your models.**

We have modified the text to discuss this (page 2, lines 7-10).

**Additional remark 2: Giving values as explained variances makes it easier to compare values between different time series as $\Delta T$ and BDC. In the moment, it is easy to compare between models in the rows of your tables, but impossible to do that between the columns of your tables.**

We have added new columns to Tables 2, 3, and 4 that show the correlation coefficient scaled by the standard deviation of the predictor time series. This is described on page 5, lines 1-3 of the manuscript, and we have modified our discussion to incorporate these values (page 5, lines 3-10 for the century regressions, and page 6, lines 32-35 for the decadal regressions). Additionally, we added a sentence to the conclusions, page 7, lines 29-30, to summarize the results.

**Additional remark 3: I have to emphasize that I am pretty sure that the trend in $\Delta T$ and in stratospheric water vapor are causally connected, I just think that a trended regression analysis is not the tool to show that. You have to avoid the impression that your trended regression analysis is a proof of that. My suggestion is the following: Add values for the explained variance of the explanatory time series (e.g. in ppm$^2$) to the tables 2, 3 and 4 (you can keep the regression coefficients or replace them by these values). Alternatively, you can add values for the regression coefficients multiplied by the standard deviations of the explanatory time series to the tables. Both explained variance and standard deviation have advantages and disadvantages: The explained variances of the explanatory timeseries add up to the overall explained variance (under the assumption that the explana-**

tory time series are uncorrelated), but values in $ppm^2$ are not very intuitive. Standard deviations are more intuitive, but don't add up. Thus, I will not give a recommendation what is better here. Next, change Figure 3 to show explained variances or standard deviations, or add an additional figure doing this. Then, base your discussion on the explained variances, where it does matter for your conclusions (e.g. in section 3.2).

We have added new columns to Tables 2, 3, and 4 that show the correlation coefficient scaled by the standard deviation of the predictor time series. This is described on page 5, lines 1-3 of the manuscript, and we have modified our discussion to incorporate these values (page 5, lines 3-10 for the century regressions, and page 6, lines 32-35 for the decadal regressions). Additionally, we added a sentence to the conclusions, page 7, lines 29-30, to summarize the results.

**Major Comment 2: It is not straightforward that more stratospheric water vapor means more warming of the troposphere, and there is not enough discussion in your paper in the moment to support your main conclusion "A stratospheric water vapor feedback exists, where a warming climate increases stratospheric water vapor, leading to further tropospheric warming". Please at least discuss the literature on that shortly [e.g. *Oinas et al.*, 2001; *Solomon et al.*, 2010]. That would give much more confidence that this statement is actually correct. Is the feedback by an increase in downward longwave radiation from the stratosphere? That does not seem to be straightforward to me. One the one hand, you have more water vapor to emit radiation. On the other hand, the stratosphere gets cooler, which reduces radiation. In a simple picture, where water vapor only emits longwave radiation and the stratosphere is heated by shortwave radiation by ozone, wouldn't the outgoing longwave radiation from a layer where you add more water vapor just stay constant to maintain radiative equilibrium, by a lowered radiative equilibrium temperature?**

We replaced the first sentence of the paper, with a sentence referring to the literature describing this process (page 1, lines 11-12), and removed the sentence in question. That said, we have not added any discussion of this to the paper because this is a well-documented phenomenon.

**Major Comment 3: It seems to me that you take the positive correlation between tropospheric temperature and stratospheric water vapor as very obvious. However, this is not simple and obvious at all. Again, discuss the literature on that shortly, and try to avoid the impression that this is an obvious fact.**

We have added a short discussion of this to the manuscript (page 2, lines 10-13)

and have hopefully changed the tone, per the reviewer's comment.

**Major Comment 4:** Since multiple regression can only show correlation but not causality, some more discussion on the supposed reasons for the correlations would be very helpful, in particular for the $\Delta T$ term. In my opinion, it should also be discussed that the reasons for a correlation can be very different in a model and in reality (i.e. based on observations). Just to give a simple example: The correlation between tropospheric temperatures and stratospheric water vapor can possibly be caused by excessive transport or diffusion of water vapor over the tropopause in the models [see *Hardiman et al.*, 2015]: Higher tropospheric temperatures means more moisture, which then could be transported by spurious vertical numerical diffusion into the stratosphere. A way to test for things like this could be e.g. to look at the tropical tropopause temperatures and their correlation to tropospheric temperatures and stratospheric water vapor.

We have added a caveat to this point on page 2, line 10.

**Major Comment 5:** Relating to this: There is a lack of information on the model performance and parameterizations of the used models. At least some information of the following list would be very helpful to assess your results. I acknowledge that it would be a lot of work to answer all of these questions for all of the models. But I think that there should be at least some discussion about how the processes in the model can affect the results. Of course, I don't want you to discuss all of these issues in detail, but to discuss things that are important for your results, i.e. take the list below as a list of suggestions.

- What is the tropopause temperature in the models, and how does it compare to measurements in terms of bias, annual cycle and trends? Can it explain the water vapor in the model or are there additional processes at work?

- How well is the Brewer-Dobson circulation represented?

- How is convection parametrized? How well does it compare to observations? Is there overshooting?

- How is radiation parametrized? What is the effect of clouds on radiation?

- What is the spatial pattern of Local dry points (LDPs) in the models and compared to reality? Can a shift in their distribution cause the correlation?

- Effect of (spurious) diffusion and transport?

Our paper is narrowly focused on quantifying the contributions of various processes to $[H_2O]_{entry}$ variability. There are many branches we could take in our discussion and we feel that we've covered the essential information required to achieve our objective. If the reviewer has a specific topic they would like to see discussed, we're happy to consider that suggestion. We also note that most of the suggestions listed above by the reviewer is already available in the literature (*Gettelman et al.* [e.g. 2010] compares TTL temperatures in the models; individual model papers discuss their parameterizations have been added to table 1 (also listed in *Morgenstern et al.* [2010, 2016]).

**Specific Comment 1: Page 1, line 1 and page 2, line 14: Please give a citation here, e.g. *Gettelman et al.* [2010] (e.g. Fig 17) or *Kim et al.* [2013]**

ACP does not prefer that citations be in the abstract, so for page 1, line 1, we will leave this to the discretion of the editor. The sentence on page 2 line 14 of the original manuscript has been removed from the current version of this manuscript.

**Specific Comment 2: Page 1, line 4: You probably mean stratospheric humidity. Please Clarify.**

Yes, "humidity" has been changed to "stratospheric humidity" in this line.

**Specific Comment 3: Page 1, line 4: In case you base that statement on your trended regression analysis, is it really correct? Correlation does not imply causality, especially in a trended regression analysis. The statement that you give on page 3, line 26-27 is a direct contradiction of what you state here. In fact, I think you cannot support that statement with the information you currently give in the paper. I would try to phrase that more carefully, e.g. by speaking of correlations, or make clear that this conclusion comes not from your trended analysis, but from some other source.**

We acknowledge that correlation does not imply causality, and we believe that we clearly base our conclusions on the detrended analysis.

**Specific Comment 4: Page 1, line 3-5: Since there is the contradicting trend from increasing cooling by the BDC (as you note here and is seen in your figure 2), can you really make the statement that the net trend in humidity is primarily driven by tropospheric warming (that would imply to me that, say, something like 80% or 90% of the net trend comes from the $\Delta$T term)? It seems to me that the trend by the BDC is in the same order of magnitude (but that the net effect of both trends is normally positive). Please add a figure showing the trends by the BDC term and the $\Delta$T term for every model to quantify**

**the trends and to underpin your statement. I think such a figure is probably easy to add.**

We added a paragraph, beginning on page 4, line 30, discussing this.

**Specific Comment 5: Page 1, line 6-7: I don't quite understand why you split your time series into 10 year chunks? Would it not be ok to compare the 100 year time series to the 10 years of observations directly?**

There are obviously many ways to compare to the MLS-based results. Our opinion is that the best way is the way we've done it in the paper. If one wants to compare the MLS results to the entire 100-year CCM run, the reader can do that by comparing the MLS coefficients (Table 4) to those from the detrended 100-year regressions (Table 3). We have added text on page 5, lines 13-14 to clarify our comparison.

**Specific Comment 6: Page 1, line 8: It is not clear to me what exactly you are referring to. Is it really that new to apply a linear regression model to these data (one of your own papers did that already: *Dessler et al.* [2013]? I suggest to delete that last sentence of the abstract or be more specific here: What is superior to what?**

We do consider this new in that we show the utility of comparison between models as a way to evaluate them. This is clearly superior to previous comparisons, *Gettelman et al.* [e.g. 2010].

**Specific Comment 7: Page 1, line 11 to Page 2, Line 2: Instead of speaking of the TTL temperatures as the determining factor, one can get more specific here. It is the temperature of the coldest point along each air mass trajectory (i.e. the Lagrangian dry point) which determines the stratospheric water vapor (except for direct injection by overshooting). In many cases this temperature will be reached at or near the tropical tropopause.**

In order to be more specific, we have modified the text on page 1, lines 16-19.

**Specific Comment 8: Page 1, line 18-19: Would be nice to add a citation here, e.g. one of the Fueglistaler papers**

We modified the text, and added several citations to support our claim regarding the Brewer-Dobson Circulation and QBO on page 2, lines 3-6.

**Specific Comment 9: Page 2, line 1-2: No, it doesn't imply that.**

See general comment 2. In addition: the local effect of more water vapor is more cooling in the stratosphere, so it is better to be more specific and to write "further tropospheric warming".

We have made that change.

**Specific Comment 10: Page 3, line 14: Probably, it is better to speak of "autocorrelation in the residuals" than of "autocorrelation of the time series", since it is only the remaining autocorrelation in the residuals that affects the uncertainty.**

We have changed "autocorrelation of the time series" to "autocorrelation in the residuals".

**Specific Comment 11: Page 3, line 18-19: You are aware that subtracting a constant does only change $\beta_0$, but does not change anything else in the regression analysis**

We know that. We believe our method is clear, as written.

**Specific Comment 12: Page 4, line 14-15: No, it doesn't confirm that, see major point 1.**

We have removed the sentence.

**Specific Comment 13: Page 4, line 18-19: This doesn't really tell you anything, see Page 4, line 14-15.**

While we agree with the reviewer's point that this should not be over-interpreted, we also feel that this is a statement worth making here. We do not believe what is written is incorrect.

**Specific Comment 14: Section 4: I don't really get the additional benefit of splitting the time series into 10 year chunks. Wouldn't a direct comparison of the observational 10 year time series and the model 100 year time series give all the information important?**

There are obviously many ways to compare to the MLS-based results. Our opinion is that the best way is the way we've done it in the paper. If one wants to compare the MLS results to the entire 100-year CCM run, the reader can do that by comparing the MLS coefficients (Table 4) to those from the detrended 100-year regressions (Table 3). We have added text on page 5, lines 13-14 to clarify our comparison.

**Specific Comment 15: Page 7, line 8-9: I find the statement that you can assess the realism of the model trend by a linear regression**

somewhat problematic. If there is a trend in stratospheric water vapor in the models and there is a trend in one of the explanatory variables, the explanatory variable will try to fit this trend, whatever the magnitude is and whatever the underlying physical reason of the trend is. If it turns out then, that the fit of the interannual variability is also good, that may give you confidence. But in general, you always have the problem that a linear regression analysis does not tell you anything about causal relationships.

We have changed the sentence to read: "We demonstrated in this paper a new way to evaluate the physical processes underlying these model trends."

**Specific Comment 16: Page 7 line 13: See second specific comment for page 1, line 4. I would phrase that more carefully.**

We don't know what the reviewer is referring to, more clarification would be appreciated.

**Specific Comment 17: Page 7, 13-16: I think it would make sense to cite some studies here and to discuss your results in comparison to other studies (briefly), e.g. studies that deal with the absence of the QBO in many models that show the influence of the BDC on tropopause temperatures and its increasing trend etc.**

We have added citations to page 7,lines 26-30; page 8 lines 1-3, that investigate influences of both the BDC and QBO on the TTL.

**Specific Comment 18: Page 7, line 21: I would agree, but I would base that statement mainly on the detrended regression analysis. If there is a good overall fit of the detrended model, you can have some confidence that the explanatory time series actually are relevant processes for the regression variable, and that the magnitude of their fit does tell you something. Since, regression analysis does not tell you anything about causal relationships however, you need to put some a priori knowledge into that. For that reason, I would be very careful to interpret the trended analysis, since there is the danger that there is no causal relationship between the trends (and the trends lead to a correlation between explanatory variables, which can make the magnitude of the fit for these variables a little bit arbitrary in the worst case.**

We agree but don't believe this is a problem, as written.

**Specific Comment 19: Page 7, line 22: That is a conclusion I would mainly draw from comparison with observations or testing the model's processes. A regression model can only help you in confirming this.**

**E.g. What would happen if all models would overestimate variability of water vapor in the future? Your fit coefficients would get larger to try to fit this variability better. Do you learn from that that the model does a good job?**

We have modified the text on page 8, lines 8-9 to account for this.

**Specific Comment 20: Page 7, line 22-23: This might however also be a deficiency of the regression approach, e.g. an explanatory variable that is no perfect proxy for the BDC, or that the trends dominate the fit (which gives rise to correlation between the explanatory variables leading to uncertainties in the magnitude of the fit for the BDC.**

It's always possible that our analysis might be wrong (for a large number of reasons), but we feel our work is adequately caveatted. If the author has a specific uncertainty/caveat that they'd like us to add, we're happy to consider it.

**Technical Revision 1: In the title you write "lower-stratospheric", later you write "lower stratospheric" would be nice to have consistency.**

"Lower stratospheric" has been changed to "lower-stratospheric" throughout the paper.

**Technical Revision 2: Page 2, line 27: Change "ozone-depleting substance" to "ozone-depleting substances"**

Done

**Technical Revision 3: Page 2, line 31: Change "described described" to "described".**

Done

**Technical Revision 4: Page 5, line 12: A period is missing ("...regression. However").**

Done

**Technical Revision 5: Page 7, line 22: Change "appear do" to "appear to do"**

Done

**References**

Dessler, A. E., M. R. Schoeberl, T. Wang, S. M. Davis, and K. H. Rosenlof (2013), Stratospheric water vapor feedback, *PNAS*, *110*(45), 18,087–18,091, doi:10.1073/pnas.1310344110.

Gettelman, A., et al. (2010), Multimodel assessment of the upper troposhere and lower stratopshere: Tropics and global trends, *J. Geophys. Res.*, *115*(D3), doi:10.1029/2009JD013638.

Hardiman, S. C., et al. (2015), Processes controlling tropical tropopause temperature and stratospheric water vapor in climate models., *J. Climate*, *28*, 6516–6535, doi:10.1175/JCLI-D-15-0075.1.

Kim, J., K. M. Grise, and S.-W. Son (2013), Thermal characteristics of the cold-point tropopause region in cmip5 models, *Journal of Geophysical Research: Atmospheres*, *118*(16), 8827–8841, doi:10.1002/jgrd.50649.

Morgenstern, O., et al. (2010), Review of the formulation of present-generation stratospheric chemistry-climate models and associated external forcings, *J. Geophys. Res.*, *115*(D3), doi:10.1029/2009JD013728.

Morgenstern, O., et al. (2016), Review of the global models used within the chemistry-climate model initiative (ccmi), *GMD*, doi:10.5194/gmd-2016-199.

Oinas, V., A. A. Lacis, D. Rind, D. T. Shindell, and J. E. Hansen (2001), Radiative cooling by stratospheric water vapor: Big differences in gcm results, *Geophysical Research Letters*, *28*(14), 2791–2794, doi:10.1029/2001GL013137.

Solomon, S., K. H. Rosenlof, R. W. Portmann, J. S. Daniel, S. M. Davis, T. J. Sanford, and P. Gian-Kasper (2010), Contributions of stratospheric water vapor to decadal changes in the rate of global warming, *Science*, *327*(5970), 1219–1223, doi:10.1126/science.1182488.

---

## Author Comment (AC2) · 20 Mar 2017

**acp reviewer 2**

**Kevin Smalley**

**March 2017**

We thank the second reviewer for their helpful remarks.

**Smalley et al. analyse CCM model predictions of stratospheric water changes over the 21st century. A multivariate linear regression is applied to the models' stratospheric water entry mixing ratios (''$[H_2O]_{entry}$''), with the explanatory variables being a ''tropospheric temperature index'', a ''Brewer Dobson strength'' index, and a QBO index; this analysis follows the method of Dessler et al. (2013). Overall, the analysis is straight- forward, and the results are clearly described. I do not comment on the aspects of the statistical analysis brought up by the other reviewer.**

**However, this reviewer cannot quite see that ''Our approach provides more insight into model processes than simply comparing $[H_2O]_{entry}$ or TTL temperatures.'' (Page 7/Line 19).**

We strongly disagree with this comment. Comparing water vapor and TTL temperatures tells you nothing about the contribution of individual processes that are responsible for $[H_2O]_{entry}$ variability. Or analysis breaks down variability in $[H_2O]_{entry}$ by process. That being said, we modified the text on page 8, lines 8-10 to try to make this clearer.

**Rather, the paper is somewhat superficial (it certainly does not help that (Page 2/Line 13): ''Finally, a warmer troposphere tends to increase $[H_2O]entry$, although whether this is through influence on TTL temperatures or some other mechanism such as convective ice lofting, is not clear.''), and results are few. It would be great if the authors would work out the connection between tropopause temperatures and $[H_2O]_{entry}$ in the models, and the connection between ''tropospheric temperature'' and tropopause temperature.**

We disagree that the paper is superficial. We view this as an important new technique to diagnose processes in CCMs, which can reveal problems in the CCMs not apparent by just looking at $[H_2O]entry$ and TTL temperatures. That said, we have added more text (lines 7-13 on page 2) that discusses the connection between tropospheric temperature, TTL temperature, and $[H_2O]_{entry}$ .

**The QBO results would also deserve some further analysis - for the 21st century analysis, annual mean data is analysed. This evidently removes much of the variance associated with the QBO, and it appears that the lack of influence of the QBO (as e.g. shown in Figure 2) is due to a lack of a trend in the QBO index. This evidently begs the question why the model does not have a QBO trend when it has been argued that the tropospheric expansion associated with global warming would have an impact on the lower stratospheric QBO - and as such would be reflected in the QBO index. While this may not have an impact on $[H_2O]entry$ (because the QBO influence at the rising tropopause level main remain constant over time), it would be useful to have some more information why the QBO index (as e.g. shown in Figure 2) does not show a trend.**

We first note that this comment shows the usefulness of our analysis (contradicting the reviewer's earlier comment): just comparing $[H_2O]entry$ and TTL temperatures would not reveal this problem with the QBO. That said, we disagree with the overall comment. Our paper is designed to understand how these processes (BDC, QBO, $\Delta T$) affect $[H_2O]entry$, not why the processes evolve as they do. Understanding why the BDC, QBO, etc. evolve as they do over the 21st century is far beyond the scope of this paper. Our paper is nonetheless an extremely useful result — by identifying this issue, our paper will spur additional research into why the QBO is not impact water vapor in the way suggested by the models.

**Two additional minor comments:**

**Please provide a reference for the statement that "Virtually all climate models ..." (page 2/Line 14)**

This sentence has been removed from the current manuscript

**and some more information about the differences in results for models that participated in CCMI-I and CCMVal-2 would be useful.**

After lengthy consideration, we've decided that there's no easy way to summarize the differences in the models in these two groups because there are no systematic differences. Trying to summarize the differences in the text therefore was unwieldy and created difficult-to-read, boring text. If people are interested in this, they can determine it using the Tables in our paper.

---

## Referee Report (RR1)

While I acknowledge that you resolved several of the issues discussed in my review and I am quite pleased now with the discussion of the linear regression, I am sorry to say that I am not willing to accept your manuscript in the current version. You ignored or misinterpreted several of my major comments (2, 3, 4, 5) and specific comments. Don't understand me wrong, you don't need to follow all of my comments, as long as you give good arguments. But that isn't the case here in my opinion. Many of my comments to the original manuscript aimed at the fact that the paper was lacking a well-balanced discussion and assessment of the results. The new manuscript version certainly has improved upon this, but there are still quite a number of issues.

**General**

- There are obvious omissions in the discussion that possibly could lead the reader to wrong conclusions (e.g. not discussing the known biases in tropopause temperature in many models or the fact that the correlation between stratospheric water vapor and tropospheric temperature can well be caused by spurious diffusion over the tropopause, see major comments 4–5).

- I noticed that you uploaded a supplement containing additional figures that add the models not shown in Figure 2. This is a very welcome addition to the paper in my opinion and I would strongly suggest to move these figures into the main body of the paper, since they add a lot of interesting information and only very moderately increase the length of your paper. If you reduce the size of the figures a little bit, they perhaps would fit on two pages as two figures with sub-panels.

- Please consider that you are addressing a broader audience here. What may seem completely obvious to you and some of your colleagues, may not be general knowledge in the wider atmospheric community.

- Just a suggestion for a title that reflects a little bit better what you have done: "Contribution of different processes to changes in tropical lower stratospheric water vapor in chemistery-climate models" or if it is ok that the title is a little bit longer "Contribution of the Brewer-Dobson circulation, the quasi-biennial oscillation and of changes in tropospheric temperature to changes in tropical lower stratospheric water vapor in chemistery-climate models".

- Partially resolved, but some of my comments in the following address this: Try to avoid exaggerations (e.g. the last sentence of the abstract), discuss other studies that are relevant in the context of your paper and don't draw conclusions that are not supported by the results of your study.

**Further discussion on old comments and your replies**

- Major comment 2: I am pleased that you removed the sentence in question and added the additional sentence to page 1, lines 11–12.

  You write "that is a well-documented phenomenon" and "water vapor is well-known to be a greenhouse gas". This is of course correct, and of course, I did not question this in any way. Nobody denies that water vapor is a greenhouse gas.

  But that is not the point here. Maybe I was not clear enough in my explanation and what I aimed at. The point is if there is a feedback on tropospheric temperatures. You need detailed radiative transfer model calculations to show that there is a significant increase in radiative forcing or temperatures of the troposphere by increases in stratospheric water vapor. None of the papers cited by me or you states a priori that there is a relevant radiative forcing of the troposphere from stratospheric water vapor. All these studies run a radiative transfer model, and then draw this as a conclusion by giving some value for the radiative forcing or temperature change. In addition, a feedback requires that higher tropospheric temperatures lead to higher tropopause temperatures, which is even less clear a priori, see major comment 3 in the original review and this document.

  The second thing is that "well-documented phenomenon" does not really hit the point. It may be well-documented by studies that are known to a certain part of our community, but you are writing for a wider audience here, which may not necessarily know the same papers as you. You can't expect the same a priori knowlegde from everyone and it is the purpose of an introduction to point the reader to the relevant literature.

  That said, I have no objections that you discuss a possible feedback here, as long as you make clear that this is not obvious and discuss the literature, and as long as you make clear that this is not a result of your study. You don't need to delete any reference to that.

- Major comment 3: You state that you have added discussion at page 2, line 10–13. But there is no discussion at this place. Did you confuse pages or line numbers here? Is it the discussion at lines 5–8?

  I am not satisfied with this. There is still no discussion that the correlation of stratospheric water vapor and tropospheric temperatures due to their long-term increase is basically a model phenomenon and can't be confirmed by the available observations. There is simply no clear trend in tropopause temperatures or water vapor in the observations (e.g. Gettelman, Fueglistaler). In addition, you don't discuss that the correlation between tropospheric warming and increasing tropopause temperatures is not that obvious from a theoretical point of view (e.g. Lin, Shepherd).

  Quite in the contrary you state "There are good physical reasons for this connection": Please rephrase. This sounds more like an annoyed comment aimed at me than as a statement aimed at the reader. And it is somewhat

ironic that you cite the Lin et al. paper here: If I may cite from the abstract: "Given the subtle nature of the balance among all these factors, it might be surprising that almost all GCMs and CCMs predict a warming [. . . ] of the tropical tropopause [. . . ]", and later (section 4) "In practice, the magnitude of tropopause warming vary vastly from model to model". I may also cite Shepherd (2002), page 778, referring to the sketch showing the conceptual relationship between tropospheric warming and warming at the tropopause "[. . . ] and certainly not as simple as depicted in Fig. 6b". And please see what I have said to specific comment 16. In particular, why don't you mention that this is only seen in models, but there is no clear evidence from observations? In summary, please try to give a more well-balanced discussion here (or in the conclusions, see specific comment 16).

- Major comments 4 and 5: I am not satisfied how you treat these major comments, which are basically ignored. I certainly do not want you to change the scope of the paper or to bloat it with unnecessary information. However, discussing the performance of the aspects of the models which are important for your analysis is crucial for the reader to be able to assess your results and their reliability (especially to assess if these model based results can be transferred to reality).

Interestingly, you discuss the QBO term in some detail, but largely avoid to discuss the $\Delta T$ term. Since you ask for specific topics that I would like to see discussed, here is one: Discuss the bias and annual cycle of tropopause temperatures *compared to observations*, in a similar manner as in Fig. 1 of Gettelman et al. (2010). It is not sufficient to point me to the Gettelman paper. It does not discuss the same models, and I am not able to find out easily if the 6 models that are discussed both in the Gettelman paper and in your paper have the same model version etc. It is also not sufficient to point me to the papers that you added to Table 1. First of all, you can't demand from the reader (or the reviewer) to read through 15 lengthy papers to find out some information that is significant for your paper. Next, by quickly scanning through the cited papers, I am pretty sure that most of them do not contain the relevant information (e.g. tropopause temperatures).

Another specific topic which is important to discuss in my opinion is spurious diffusion of water vapor across the tropopause. There is an extremely large gradient of water vapor near the tropopause and at the same time, models are well known to be too diffusive compared to reality (especially in the stratosphere, where the effective diffusion coefficients are 100 times smaller than in the troposphere). This problem is well documented in the literature (e.g. Gettelman et al., 2010, page 11, Hardiman, 2015, section 3). It is well possible that the relationship between stratospheric water vapor and tropospheric temperature is dominated by this effect (at least in some models) and not discussing this may lead the reader to the wrong

conclusion that he can transfer the results of your study (trends, contribution of the different terms) more easily to the real behaviour of the atmosphere than it is actually the case.

In this respect, I am also not very satisfied with your answer to major comment 4. You say that you have added a caveat to the paper, but in fact you did not adress the point I discussed in major comment 4. I was talking about spurious diffusion in the comment, but the caveat you added to the text deals with overshooting convection. This is certainly also an interesting point, but not what I was talking about.

Another issue is the BDC, which is also not discussed. How well the BDC is represented in the models will have implications for the contribution of the BDC to the trend and variability of stratospheric water vapor in your regression model. E.g., if the BDC is too fast in a model (compared e.g. to $w^*$ derived from reanalyses), it will lead to an overestimation of this term in your regression analysis compared to reality.

- Specific comment 1 (was Page 1 , Line 1 and Page 2, line 14): Was there any reason apart from this comment that caused you to remove the sentence? The aim of my comment was certainly not that you remove the sentence, but that you add the citations. Now there is the unfortunate situation that the sentence is still in the manuscript (in the abstract), but that you can't give the relevant citations (I acknowledge that it is no good idea to cite in the abstract). And to cite the relevant literature is certainly appropriate for this central statement.

- Specific comment 3: For the reasons given in my review, I still think this is a problematic statement. In addition: In your reply to this comment, you state "we clearly base our conclusions on the detrended analysis". But this sentence explicitely refers to the trend in humidity. What do you want to tell me with your statement? Please also see my detailed comments to specific comment 16 below.

- Specific comment 6 (was Page 1, line 8): I am not satisfied how you treat this comment. You neither deleted or changed the sentence, nor did you explain to me in your reply what you mean by "superior" and to what the statement refers in a satisfactory way.

This comment was one of the more important specific comments I made, since this statement is in the abstract at a rather prominent position, and it is just an unproven und unclear statement. You should try to avoid the impression that you put this sentence into the abstract just to create interest for your article, without anything really supporting this statement.

Since you refer to the Gettelman paper in your reply: Do you mean that applying a multiple linear regression model is better than just looking at plots of stratospheric humidity and tropopause temperature? Then, why don't you write it, neither in the reply to my comments, nor in the abstract?

And if this is really what you mean, is it really worth mentioning? It was certainly not the intention of Gettelman et al. to do a multiple regression analysis and for the purpose of their paper, it was sufficient to show the plots. And there are studies, including your own studies, which already used multiple linear regression. So, what is the point here?

- Specific comment 7 (was Page 1, line 11): It is nice that you refer to the LDPs now, but unfortunately, the sentence is not quite correct. The coldest temperatures in the TTL are not necessarily at the location where an individual trajectory has its LDP, which may cross the tropopause at a warmer location. I suggest to rephrase the sentence so that the statement is correct.

- Specific comment 16 (was Page 7, line 13, now lines 26–27): That is referring to the identical sentence on page 1, line 4 (old manuscript) and the comment referring to it (specific comment 3). There needs to be more discussion here, and I find the statement here problematic. You can't draw the conclusion that the trend in the warming of the troposphere drives the trend in stratospheric water vapor from your trended regression analysis (as you admit in line 26–27, page 3 in the old manuscript). *Any* timeseries with a trend will fit your stratospheric water vapor time series. I.e., it is just not correct to say "we find". I suggest to change the sentence to "We find that in our trended regression analysis, the trend in stratospheric water vapor is explained largely by the trend in tropospheric temperature." That has a completely different meaning, in particular, it does not imply that the change in tropospheric temperature is the indisputable underlying reason for the trend in stratospheric humidity in the models. In addition, it does not imply that in reality, a trend in tropospheric temperature will imply a trend in stratospheric humidity. I am aware that you write "in the CCMs" in this sentence, but there is no discussion in the paper that the trend in stratospheric humidity and in tropopause temperature are basically a model phenomenon. The observations of water vapor and temperature do not support this conclusion clearly in the moment. In addition, it is also not a priori clear from a theoretical point of view. See my major comment 3 of the original review again for this.

**New comments**

- Page 1, line 2: Better: "We analyze the trend and variability [. . .]". Without interannual variability in at least some of the variables, you would not be able to fit the explanatory time series without ambiguity to the water vapor time series (i.e. if all variables would only contain a trend, the error bars would go to infinity and the fitted values would be arbitrary).

- Page 1, line 7: "Many of the CCMs [. . .]". Rephrase or delete: a) This is an unproven statement, in particular since you explicitly refuse to give information about model performance in this paper. b) This is far too generic. Models may perform well in some variables, but no so good in

others, and this will also vary from model to model. Be more specific. c) It is unclear what observations you are referring to. d) In particular referring to the trends in water vapor and tropopause temperature: This is a particularly bad example for a "credible" prediction. It is unclear from observations and theory, and is mainly based on the belief that the models do model these particular aspects of the climate system well.

- Page 1, lines 11–12: Please write "increasing it will lead to additional warming of the troposphere" and not "of the climate system". That is too generic. More stratospheric water vapor cools the stratosphere, so this statement is obviously not quite correct.

- Page 1, lines 11–12: "Stratospheric water vapor is a greenhouse gas". Change that to "Water vapor is a greenhouse gas". If a gas is a greenhouse gas or not does not depend on the altitude. It is defined as a gas absorbing in the thermal infrared. And then start a new sentence "Increasing stratospheric water vapor will lead to additional warming of the troposphere, as shown by [citations]"

- Page 2, line 8: Does the correlation of 0.91 refer to the trended or detrended variables? It would be really helpful for your argumentation if the interannual changes would be correlated.

- Page 2, line 19: Better "worked well in reproducing trend and variability"? It is no surprise that it is easy to fit a variable with a trend to another variable with a trend.

- Page 2, line 22: What do you mean by comparison to observations? Do you mean to apply the same regression model to time series of observations and to compare the results?

- Page 2, line 23: Here applies the same comment that I had to Page 7, line 8–9 (original manuscript, specific comment 15). This is solved in the conclusions now, but not here.

- Page 3, line 25: Don't exaggerate. Can we agree on "good job"?

- Page 4, line 1–2. Half of the models shows an explained variance decreased by more than 0.2. That is not "slightly" smaller. Suggestion: "moderately".

- Page 4, line 32 to Page 5, 4: The term "standardized regression coefficient" is a little bit unfortunate. It confused me several times when reading this section, because it suggests something different than actually intended. This is not a regression coefficient, but something like a "variability of the fitted time series" or "standard deviation of the fitted time series" or "square root of the explained variance". Please change.

- Same paragraph: I noticed that in several of the models (e.g. CMAM-CCMI, GEOSCCM, GEOSCCM-CCMI), the variability in the stratospheric water vapor time series mostly comes from the variability in BDC and QBO, with almost no variability in the $\Delta T$ time series. That means that the magnitude of the fitted trend in $\Delta T$ is very dependent on the magnitude of the *interannual variability* of the QBO/BDC in these models, since the $\Delta T$ term, which is almost a pure trend, will fit "what is left from the trend" after matching the interannual variability and trend of the QBO/BDC time series. This may be worth mentioning, since this is a good example of an effect on the $\Delta T$ trend which is not "physical", but "numerical".

- Page 7, line 21: "A new way"? See specific comment 6.

**Technical comments**

- The citations Scinocca et al. (2008a) and Scinocca et al. (2008b) are exactly identical.

- Tables 2, 3, 4: Since $\mathrm{STD}(\beta_{\Delta T}\Delta T) = |\beta_{\Delta T}|\mathrm{STD}(\Delta T)$, I suggest to omit the negative sign in all columns showing standard deviations.

---

## Author Response (AR3)

**Reply to Reviewer #1**

We thank Reviewer 1 for their continued concern with our manuscript. In our response, the reviewer's comments are bolded, our answers are normal weight, and anything that we change in the paper is italicized.

**1  general**

There are obvious omissions in the discussion that possibly could lead the reader to wrong conclusions (e.g. not discussing the known biases in tropopause temperature in many models or the fact that the correlation between stratospheric water vapor and tropospheric temperature can well be caused by spurious diffusion over the tropopause, see major comments 4–5).

**We hope that we have alleviated this problem with our draft, see answers to other comments.**

I noticed that you uploaded a supplement containing additional figures that add the models not shown in Figure 2. This is a very welcome addition to the paper in my opinion and I would strongly suggest to move these figures into the main body of the paper, since they add a lot of interesting information and only very moderately increase the length of your paper. If you reduce the size of the figures a little bit, they perhaps would fit on two pages as two figures with sub-panels.

**The reviewer makes a good point that the supplement is not well referenced in the paper. To address that, we have added a sentence to page 3, lines 20-21 to more clearly reference the supplement, as well as a short paragraph in section 3.2, page 5 lines 6-9. We do not see sufficient value in moving these figures into the main text of the paper to do that, so we'll leave them in the supplement.**

Please consider that you are addressing a broader audience here. What may seem completely obvious to you and some of your colleagues, may not be general knowledge in the wider atmospheric community.

**We hope that we have alleviated this problem with our draft, see answers to other comments.**

Just a suggestion for a title that reflects a little bit better what you have done: "Contribution of different processes to changes in tropical lower stratospheric water vapor in chemistry-climate models" or if it is ok that the title is a little bit longer "Contribution of the Brewer-Dobson circulation, the quasi-biennial oscillation and of changes in tropospheric temperature to changes in tropical lower stratospheric water vapor in chemistry-climate models".

**The Title has been changed to** *"Contribution of different processes to changes in tropical lower stratospheric water vapor in chemistry-climate models"*

Partially resolved, but some of my comments in the following address this: Try to avoid exaggerations (e.g. the last sentence of the abstract), discuss other studies that are relevant in the context of your paper and don't draw conclusions that are not supported by the results of your study.

**We hope that we have alleviated this problem with our draft, see answers to other comments.**

**2  Further discussion on old comments and your replies**

Major comment 2: I am pleased that you removed the sentence in question and added the additional sentence to page 1, lines 11–12. You write "that is a well-documented phenomenon" and "water vapor is well-known to be a greenhouse gas". This is of

course, correct, and of course, I did not question this in any way. Nobody denies that water vapor is a greenhouse gas. But that is not the point here. Maybe I was not clear enough in my explanation and what I aimed at. The point is if there is a feedback on tropospheric temperatures. You need detailed radiative transfer model calculations to show that there is a significant increase in radiative forcing or temperatures of the troposphere by increases in stratospheric water vapor. None of the papers cited by me or you states a priori that there is a relevant radiative forcing of the troposphere from stratospheric water vapor. All these studies run a radiative transfer model, and then draw this as a conclusion by giving some value for the radiative forcing or temperature change. In addition, a feedback requires that higher tropospheric temperatures lead to higher tropopause temperatures, which is even less clear a priori, see major comment 3 in the original review and this document. The second thing is that "well-documented phenomenon" does not really hit the point. It may be well-documented by studies that are known to a certain part of our community, but you are writing for a wider audience here, which may not necessarily know the same papers as you. You can't expect the same a priori knowledge from everyone and it is the purpose of an introduction to point the reader to the relevant literature. That said, I have no objections that you discuss a possible feedback here, as long as you make clear that this is not obvious and discuss the literature, and as long as you make clear that this is not a result of your study. You don't need to delete any reference to that.

**This material is all well known by the expert community. We disagree with the assertion from the reviewer that this paper should be written for a wide audience. Our view is that ACP is read by specialists who are familiar with the literature. Expecting us to write this paper for truly broad audience (e.g., Rev. Geophys, BAMS, EOS) would radically change the nature of the paper and we don't feel that's appropriate.**

Major comment 3: You state that you have added discussion at page 2, line 10–13. But there is no discussion at this place. Did you confuse pages or line numbers here? Is it the discussion at lines 5–8? I am not satisfied with this. There is still no discussion that the correlation of stratospheric water vapor and tropospheric temperatures due to their long-term increase is basically a model phenomenon and can't be confirmed by the available observations. There is simply no clear trend in tropopause temperatures or water vapor in the observations (e.g. Get- telman, Fueglistaler). In addition, you don't discuss that the correlation between tropospheric warming and increasing tropopause temperatures is not that obvious from a theoretical point of view (e.g. Lin, Shepherd). Quite in the contrary you state "There are good physical reasons for this connection": Please rephrase. This sounds more like an annoyed comment aimed at me than as a statement aimed at the reader. And it is somewhat ironic that you cite the Lin et al. paper here: If I may cite from the abstract: "Given the subtle nature of the balance among all these factors, it might be surprising that almost all GCMs and CCMs predict a warming [. . . ] of the tropical tropopause [. . . ]", and later (section 4) "In practice, the magnitude of tropopause warming vary vastly from model to model". I may also cite Shepherd (2002), page 778, referring to the sketch showing the conceptual relationship between tropospheric warming and warming at the tropopause "[...] and certainly not as simple as depicted in Fig. 6b". And please see what I have said to specific comment 16. In particular, why don't you mention that this is only seen in models, but there is no clear evidence from observations? In summary, please try to give a more well-balanced discussion here (or in the conclusions, see specific comment 16).

**We should have been more specific. Additionally, we modified our discussion of the relationship between tropospheric warming and TTL temperatures (see page 2, lines 5-7)**

Major comments 4 and 5: I am not satisfied how you treat these major comments, which are basically ignored. I certainly do not want you to change the scope of the paper or to bloat it with unnecessary information. However, discussing the performance of the aspects of the models which are important for your analysis is crucial for the reader to be able to assess your results and their reliability (especially to assess if these model based results can be transferred to reality).

Interestingly, you discuss the QBO term in some detail, but largely avoid to discuss the $\Delta T$ term. Since you ask for specific topics that I would like to see discussed, here is one: Discuss the bias and annual cycle of tropopause temperatures compared to observations, in a similar manner as in Fig. 1 of Gettelman et al. (2010). It is not sufficient to point me to the Gettelman paper. It does not discuss the same models, and I am not able to find out easily if the 6 models that are discussed both in the Gettelman paper and in your paper have the same model version etc. It is also not sufficient to point me to the papers that you

added to Table 1. First of all, you can't demand from the reader (or the reviewer) to read through 15 lengthy papers to find out some information that is significant for your paper. Next, by quickly scanning through the cited papers, I am pretty sure that most of them do not contain the relevant information (e.g. tropopause temperatures).

5 **We discuss the QBO term in depth because that's the one where the models do the worst, so it's obviously of most interest. We don't discuss biases in the annual cycle of temperature because that seems entirely irrelevant to this analysis. The first thing we do in our analysis is either average over the annual cycle or remove it. So it's not clear how biases in it will impact the paper. In addition, the goal of our paper is to determine the response of $H_2O$ to changes in the temperature. The temperature doesn't have to be right for the response to be right; nor is the response right if we show**
10 **that the temperatures are right. These are completely separate quantities.**

Another specific topic which is important to discuss in my opinion is spurious diffusion of water vapor across the tropopause. There is an extremely large gradient of water vapor near the tropopause and at the same time, models are well known to be too diffusive compared to reality (especially in the stratosphere, where the effective diffusion coefficients are 100 times smaller
15 than in the troposphere). This problem is well documented in the literature (e.g. Gettelman et al., 2010, page 11, Hardiman, 2015, section 3). It is well possible that the relationship between stratospheric water vapor and tropospheric temperature is dominated by this effect (at least in some models) and not discussing this may lead the reader to the wrong conclusion that he can transfer the results of your study (trends, contribution of the different terms) more easily to the real behavior of the atmosphere than it is actually the case.
20
**We added a caveat to page 4, lines 22-26.**

In this respect, I am also not very satisfied with your answer to major comment 4. You say that you have added a caveat to the paper, but in fact you did not address the point I discussed in major comment 4. I was talking about spurious diffusion
25 in the comment, but the caveat you added to the text deals with overshooting convection. This is certainly also an interesting point, but not what I was talking about.

**We added a caveat to page 4, lines 22-26.**

30 Another issue is the BDC, which is also not discussed. How well the BDC is represented in the models will have implications for the contribution of the BDC to the trend and variability of stratospheric water vapor in your regression model. E.g., if the BDC is too fast in a model (compared e.g. to w∗ derived from reanalyses), it will lead to an overestimation of this term in your regression analysis compared to reality.

35 **We added a new discussion to page 3, lines 4-11. That said, we disagree with the premise of this comment. Our analysis measures the response of H2O to changes in the BDC. The BDC doesn't have to be right for the response to be right; nor is the response right if we show that the BDC is right. These are completely separate quantities.**

Specific comment 1 (was Page 1 , Line 1 and Page 2, line 14): Was there any reason apart from this comment that caused
40 you to remove the sentence? The aim of my comment was certainly not that you remove the sentence, but that you add the citations. Now there is the unfortunate situation that the sentence is still in the manuscript (in the abstract), but that you can't give the relevant citations (I acknowledge that it is no good idea to cite in the abstract). And to cite the relevant literature is certainly appropriate for this central statement.

45 **We removed the sentence originally on page 2, line 14, because it didn't fit where it was written. Additionally, we modified the first sentence of the abstract to remove the need for a citation; see page 1, line 1.**

Specific comment 3: For the reasons given in my review, I still think this is a problematic statement. In addition: In your reply to this comment, you state "we clearly base our conclusions on the detrended analysis". But this sentence explicitly

refs to the trend in humidity. What do you want to tell me with your statement? Please also see my detailed comments to specific comment 16 below.

**We understand that correlation does not imply causality, and that the trended century regressions are not, by themselves, convincing. But we feel that the totality of the work — trended regressions, detrended regressions, and decadal regressions — provides sufficient evidence to make this statement. As a result, we've left this as is.**

Specific comment 6 (was Page 1, line 8): I am not satisfied how you treat this comment. You neither deleted or changed the sentence, nor did you explain to me in your reply what you mean by "superior" and to what the statement refers in a satisfactory way. This comment was one of the more important specific comments I made, since this statement is in the abstract at a rather prominent position, and it is just an unproven and unclear statement. You should try to avoid the impression that you put this sentence into the abstract just to create interest for your article, without anything really supporting this statement. Since you refer to the Gettelman paper in your reply: Do you mean that applying a multiple linear regression model is better than just looking at plots of stratospheric humidity and tropopause temperature? Then, why don't you write it, neither in the reply to my comments, nor in the abstract? And if this is really what you mean, is it really worth mentioning? It was certainly not the intention of Gettelman et al. to do a multiple regression analysis and for the purpose of their paper, it was sufficient to show the plots. And there are studies, including your own studies, which already used multiple linear regression. So, what is the point here?

**We disagree with this comment. We feel this is a new and novel way to look at models' regulation of stratospheric water vapor. Obviously, readers will render the final verdict.**

Specific comment 7 (was Page 1, line 11): It is nice that you refer to the LDPs now, but unfortunately, the sentence is not quite correct. The coldest temperatures in the TTL are not necessarily at the location where an individual trajectory has its LDP, which may cross the tropopause at a warmer location. I suggest to rephrase the sentence so that the statement is correct.

**This sentence has been modified; see page 1, lines 16-18.**

Specific comment 16 (was Page 7, line 13, now lines 26–27): That is referring to the identical sentence on page 1, line 4 (old manuscript) and the comment referring to it (specific comment 3). There needs to be more discussion here, and I find the statement here problematic. You can't draw the conclusion that the trend in the warming of the troposphere drives the trend in stratospheric water vapor from your trended regression analysis (as you admit in line 26–27, page 3 in the old manuscript). Any timeseries with a trend will fit your stratospheric water vapor time series. I.e., it is just not correct to say "we find". I suggest to change the sentence to "We find that in our trended regression analysis, the trend in stratospheric water vapor is explained largely by the trend in tropospheric temperature." That has a completely different meaning, in particular, it does not imply that the change in tropospheric temperature is the indisputable under- lying reason for the trend in stratospheric humidity in the models. In addition, it does not imply that in reality, a trend in tropospheric temperature will imply a trend in stratospheric humidity. I am aware that you write "in the CCMs" in this sentence, but there is no discussion in the paper that the trend in stratospheric humidity and in tropopause temperature are basically a model phenomenon. The observations of water vapor and temperature do not support this conclusion clearly in the moment. In addition, it is also not a priori clear from a theoretical point of view. See my major comment 3 of the original review again for this.

**See new discussion on page 3, lines 25-28.**

**3 New Comments**

Page 1, line 2: Better: "We analyze the trend and variability [. . . ]". With- out interannual variability in at least some of the variables, you would not be able to fit the explanatory time series without ambiguity to the water vapor time series (i.e. if all variables would only contain a trend, the error bars would go to infinity and the fitted values would be arbitrary).

**We amended this sentence (see page 1, lines 1-3).**

Page 1, line 7: "Many of the CCMs [. . . ]". Rephrase or delete: a) This is an unproven statement, in particular since you explicitly refuse to give information about model performance in this paper. b) This is far too generic. Models may perform well in some variables, but no so good in others, and this will also vary from model to model. Be more specific. c) It is unclear what observations you are referring to. d) In particular referring to the trends in water vapor and tropopause temperature: This is a particularly bad example for a "credible" prediction. It is unclear from observations and theory, and is mainly based on the belief that the models do model these particular aspects of the climate system well.

**We have modified the sentence to explicitly say that we're talking about the performance in the regression. While the author is correct that this sentence does not give all of the details, it is located in the abstract so a general overview is most appropriate. More details about the comparisons are found in the text (starting on line 22 of page 5).**

Page 1, lines 11–12: Please write "increasing it will lead to additional warming of the troposphere" and not "of the climate sys- tem". That is too generic. More stratospheric water vapor cools the stratosphere, so this statement is obviously not quite correct.

**In response to a different comment, we've removed this sentence.**

Page 1, lines 11–12: "Stratospheric water vapor is a greenhouse gas". Change that to "Water vapor is a greenhouse gas". If a gas is a greenhouse gas or not does not depend on the altitude. It is defined as a gas absorbing in the thermal infrared. And then start a new sentence "Increasing stratospheric water vapor will lead to additional warming of the troposphere, as shown by [citations]"

**While we agree with the reviewer that water vapor is generally a greenhouse gas, the phrasing of our statement is in our opinion completely clear. We've therefore left this sentence as-is.**

Page 2, line 8: Does the correlation of 0.91 refer to the trended or de- trended variables? It would be really helpful for your argumentation if the interannual changes would be correlated.

**In light of the previous comments and our responses, we believe that this point has been sufficiently litigated. For these reasons, we have removed the reference to the 0.91 correlation between TTL temperatures and tropospheric warming.**

Page 2, line 19: Better "worked well in reproducing trend and variability"? It is no surprise that it is easy to fit a variable with a trend to another variable with a trend.

**We feel our phrasing is equivalent and have left this sentence as it was in the previous version.**

Page 2, line 22: What do you mean by comparison to observations? Do you mean to apply the same regression model to time series of observations and to compare the results?

**Yes, this is what we mean. We feel this is completely clear as written.**

Page 2, line 23: Here applies the same comment that I had to Page 7, line 8–9 (original manuscript, specific comment 15).

This is solved in the conclusions now, but not here.

**We have edited this (page 2, lines 20-21), but we also think this point exaggerates our claims. We are not evaluating the models with just a linear regression — the regression is based on well-known and understood physics, so the ability to reproduce the linear regression DOES tell us something about the quality of the fit.**

Page 3, line 25: Don't exaggerate. Can we agree on "good job"?

*"Excellent"* **has been changed to** *"good job"*

Page 4, line 1–2. Half of the models shows an explained variance decreased by more than 0.2. That is not "slightly" smaller. Suggestion: "moderately".

*"slightly"* **Has been changed to** *"moderately"*

Page 4, line 32 to Page 5, 4: The term "standardized regression coefficient" is a little bit unfortunate. It confused me several times when reading this section, because it suggests something different than actually intended. This is not a regression coefficient, but something like a "variability of the fitted time series" or "standard deviation of the fitted time series" or "square root of the explained variance". Please change.

*"standardized regression coefficient"* **has been changed to** *"regression coefficient using standardized variables"*

Same paragraph: I noticed that in several of the models (e.g. CMAM- CCMI, GEOSCCM, GEOSCCM-CCMI), the variability in the stratospheric water vapor time series mostly comes from the variability in BDC and QBO, with almost no variability in the $\Delta T$ time series. That means that the magnitude of the fitted trend in $\Delta T$ is very dependent on the magnitude of the interannual variability of the QBO/BDC in these models, since the $\Delta T$ term, which is almost a pure trend, will fit "what is left from the trend" after matching the interannual variability and trend of the QBO/BDC time series. This may be worth mentioning, since this is a good example of an effect on the $\Delta T$ trend which is not "physical", but "numerical".

**We added a short paragraph to the end of page 5 lines 6-9.**

Page 7, line 21: "A new way"? See specific comment 6.

**We modified this sentence, see page 7 line 29 - page 8 line 1.**

**Reply to Reviewer #2**

In the revised version, Smalley et al. have addressed some issues raised by reviewers. However, the authors have decided to ignore the suggestion that more in-depth analysis would make the paper stronger. It is repeatedly stated that the linear regression shown in this paper is superior to "simply comparing $[H_2O]_{entry}$ to observations"; these statements make little sense to this reviewer (obviously the questions posed are different), and also does not help to make the paper stronger (I recommend deleting these sentences).

**We hope that we have alleviated this problem. But, in reference to your example (assuming the last sentence of the abstract), we disagree with this comment. We feel this is a new and novel way to look at models' regulation of stratospheric water vapor. Obviously, readers will render the final verdict..**

My main concern remains that a linear regression is useful to detect correlations and common properties, but not more. In my opinion the paper overstates the results when claiming that the linear regressions shown provide insights into processes.

**We agree that a linear regression by itself does not prove causation. But a linear regression combined with physics can be used to evaluate the relationship between each process and $[H_2O]_{entry}$. In this case, there are good physical reasons to believe that the terms in our linear regression affect $[H_2O]_{entry}$, as well as observational evidence to support each one. We reference previous literature demonstrating this in our paper. Thus, we disagree that we have overstated the value of our linear regression.**

Case in point is the Brewer-Dobson result where some models have a positive BDC coefficient (CNRM-CM5-3, NIWA-UKCA; p6/L29: All pages/lines refer to the manuscript version with changes highlighted). The paper does not explain to the reader why the authors expect the coefficient to be negative, and it would have been easy to check a few model fields as to why the coefficient in these models is positive. A brief mechanistic discussion (there's plenty of papers discussing the Newtonian cooling and ozone/other tracers that can be used for reference) would be helpful.

**We have added a new discussion to the conclusions (see page 8, lines 16 - 22) discussing the physical mechanism that connects BDC variability to the TTL.**

The revised manuscript has a supplement with 13 figures with proper caption but no text whatsoever. I could spot only one reference to the supplement in the manuscript (p3/L13). Please integrate the supplement better with the manuscript, and provide a very brief description (along with a title like "Supplementary Material for ...") in the supplement.

**We added a sentence to clearly reference the supplemental material on page 3, lines 20-21, as well as a reference to the supplement in the section 3.2 page 5 lines 6-9, and added a title to the supplemental material.**

Finally, the text deserves some careful checking; for example the first sentence of the introduction ("... so increasing it will lead ...") or the last sentence of Section 3.1 ("... show something similar ...").

**Done**

[revised manuscript text omitted]